# Organoids transplantation attenuates intestinal ischemia/reperfusion injury in mice through L-Malic acid-mediated M2 macrophage polarization

Fang-Ling Zhang[1,2], Zhen Hu[1,2], Yi-Fan Wang[1,2], Wen-Juan Zhang[1], Bo-Wei Zhou[1], Qi-Shun Sun[1], Ze-Bin Lin [1] & Ke-Xuan Liu [1]✉

Intestinal organoid transplantation is a promising therapy for the treatment of mucosal injury. However, how the transplanted organoids regulate the immune microenvironment of recipient mice and their role in treating intestinal ischemia-reperfusion (I/R) injury remains unclear. Here, we establish a method for transplanting intestinal organoids into intestinal I/R mice. We find that transplantation improve mouse survival, promote self-renewal of intestinal stem cells and regulate the immune microenvironment after intestinal I/R, depending on the enhanced ability of macrophages polarized to an anti-inflammatory M2 phenotype. Specifically, we report that L-Malic acid (MA) is highly expressed and enriched in the organoids-derived conditioned medium and cecal contents of transplanted mice, demonstrating that organoids secrete MA during engraftment. Both in vivo and in vitro experiments demonstrate that MA induces M2 macrophage polarization and restores interleukin-10 levels in a SOCS2-dependent manner. This study provides a therapeutic strategy for intestinal I/R injury.

The role of the small intestine in digestion and macronutrient absorption is well established, indicating that normal small intestine function is essential for survival. Intestinal ischemia/reperfusion (I/R) injury is a common clinical pathophysiological process that causes massive necrosis of the mucosal epithelium and the elimination of villi and crypts. This results in intestinal barrier dysfunction, which provokes deleterious complications, leading to mortality as high as 50-90%[1,2]. Intestinal I/R injury occurs in acute and chronic intestinal obstruction, acute mesenteric ischemia, cardiopulmonary bypass, abdominal aortic aneurysm surgery, small-bowel transplantation, and neonatal necrotizing enterocolitis[3]. Several biological and surgical prevention approaches have been developed, including treatment with *Lactobacillus murinus* and its metabolites[4], ischemic preconditioning[5], in attempt to help prevent intestinal I/R injury.

However, new treatment strategies that treat severe small intestinal mucosal ulcers and promote membrane repair after ischemia are needed.

Organoids are derived from self-assembling stem cells and maintain the in vivo physiological structures, functions, and genetic signatures of their original tissues[6]. Thus, organoid technology has been rapidly applied to understand stem cell biology, cancer, biochemistry, and molecular biology[7]. Previous stem cell-based therapies have shown limitations including transcriptional aberrations and low viability while used in transplantation[8]. Recently, organoids have emerged as prominent candidates for transplantation. For example, islet organoids transplantation preserves their function of secreting insulin, thus reversing diabetes in mice[9]. Moreover, ileum-derived organoids xenografted into the colon reconstituted their original

[1]Department of Anesthesiology, Nanfang Hospital, Southern Medical University, Guangzhou 510515, China. [2]These authors contributed equally: Fang-Ling Zhang, Zhen Hu, Yi-Fan Wang. ✉e-mail: liukexuan705@163.com

cholesterol absorption function and generated a functional small-intestinal colon[10]. Intestinal organoids are considered new therapeutic options for intestinal mucosal injuries. Rodent colonic organoids and small intestinal crypt-derived organoids have been used to treat homogeneous models of colonic ulcers[11,12]. Organoids derived from the human small intestine[13], colon tissue[14], and pluripotent stem cells[15] have induced the proliferation, repair, and reconstruction of healthy mucosa epithelial cells after transplantation in mice. This indicates that delivery of small intestinal organoids into injured mucosa is a promising approach for treating mucosal injury and its complications. However, whether organoids transplantation can be used as a new therapeutic method to reduce intestinal I/R-induced intestinal injury has not yet been studied and its specific underlying mechanism remains unclear.

The intestinal mucosal immune microenvironment plays a vital role in the proliferation of intestinal stem cells (ISCs)[16,17] as well as the repair of intestinal epithelial injury[18]. Immune effectors, including cytokines, growth factors, chemokines, and many other mediators are involved in the recovery of intestinal mucosal injury[19]. Several types of immune cells constitute the immune microenvironment, among which macrophages are the predominant type[20]. Macrophages functionally polarize towards an M2-like phenotype not only serve as an anti-inflammatory phenotype[20] but also actively support the self-renewal of ISCs[21]. Various metabolites produced by cells regulate the polarization of macrophages[22]. Meanwhile, intestinal organoids contribute to the production of variable anions, fluids[23], proteins, hormones[24], and extracellular vesicles[25]. Moreover, the potential effects of cell therapy in modeling the immune system have been widely reported[26,27]. However, it remains unclear whether organoids play a role in mitigating intestinal injury by secreting metabolic factors to polarize macrophages and regulate the immune microenvironment.

Here, we developed an efficient in vivo engraftment model of mouse small intestinal organoids in intestinal I/R injury to investigate the potential of organoid transplantation in promoting mucosal injury recovery. Furthermore, we discuss whether the transplanted organoids played a protective role by regulating the self-renewal of intestinal stem cells (ISCs) as well as regulating the immune microenvironment of recipient mice. We further explored the underlying mechanism in mouse intestinal I/R injury models.

## Results

### Organoids transplantation alleviates intestinal I/R injury

To determine whether transplanted organoids regulate intestinal homeostasis and promote intestinal injury recovery, we induced intestinal I/R injury in wild-type (WT) mice. First, to investigate the effects of organoids transplantation during different stages of intestinal inflammation, the mice were subjected to 50 min of ischemia and then transplanted with organoids or control solution and sacrificed at indicated time points (Fig. 1A). Next, we assessed the effect of organoids engraftment on mouse survival seven days after intestinal I/R injury. Transplantation of the organoids significantly reduced the mortality compared with that in the control group (Fig. 1B). We further confirmed the successful engraftment of organoids through immunofluorescence which revealed the organoids grown from Lgr5-eGFP-IRES-CreERT2 were attached and incorporated into the injured epithelial mucosa (Fig. 1C). Notably, our results are consistent with previous study[11] of higher engraftment rate of cultured organoids, rather than freshly isolated crypt or sorted leucine-rich repeat-containing G-protein coupled receptor 5 (Lgr5)+ stem cells 36 h after intestinal I/R injury (Supplementary Fig. 1A–D). Compared with the transplantation group, the control group exhibited worse mucosal damage with significant epithelial necrosis, mucosal villus breakdown, and ulcers at different time points before 48 h (Supplementary Fig. 2A). Scattered ulcers in the small intestine caused approximately half of the crypts to disappear in both the transplantation and control groups 6 h after

surgery (Fig. 1D). Notably, the crypt depth in the transplantation group was significantly higher than that of the control group 36 h after surgery (Fig. 1D), suggesting the successful reconstitution of the recipient's intestinal crypt after organoids were transplanted. The pathology scores of the transplantation group at 36 h were also significantly lower than those of the control group (Fig. 1E). Next, we assessed Occludin and ZO-1 levels to evaluate intestinal integrity and found that both protein and mRNA levels of Occludin and ZO-1 were higher in the transplantation group than those in the control group (Supplementary Fig. 2B–E).

Inflammatory cytokines levels fluctuated dramatically after I/R injury. Notably, the secretion of interleukin-10 (IL-10) exhibited a reverse trend over time. IL-10 expression in the transplantation group increased significantly 36 h after surgery, whereas that of the control group showed little change during 48 h after surgery (Fig. 1F, G). The production of both interleukin-6 (IL-6) and interleukin-1β (IL-1β) in the transplantation group was reduced at 36 h compared with that in the control group (Fig. 1H). These data indicated that organoids transplantation attenuated intestinal inflammation following I/R injury.

To further evaluate the ability of transplanted organoids in promoting regeneration, we found that the organoid-transplanted mice exhibited a significant increase in Ki-67+ proliferating cells at 24 and 36 h compared with that in the control group (Fig. 1I, J). Additionally, the small intestinal crypts of transplanted mice exhibited significantly more Lysozyme+ Paneth cells, olfactomedin 4 (OLFM4)+ stem cells, and Muc2+ goblet cells at 36 h, than those of control mice (Fig. 1I, K, and Supplementary Fig. 2F, G). Relative mRNA levels of Ki-67 and lysozyme in the transplantation group were drastically increased at 36 h compared with those in the control group (Fig. 1L, M). These results indicated that organoids transplantation promotes the self-renewal of ISCs.

### The myeloid landscape is essential for organoid transplantation therapy

We further examined the effects of organoids transplantation on the local mucosal immune microenvironment during and after intestinal I/R injury. Initially, massive infiltration of neutrophils and monocytes was detected at 6 h, whereas macrophages dramatically increased at 12 h (Fig. 2A). Although the proportions of neutrophils, monocytes, and macrophages increased comparably in the ischemic mucosa of both transplantation and control groups until 24 h, the increase was more significant in control mice compared with transplanted mice at 36 h, reflecting a general and maintained pro-inflammatory state in the control group. Interestingly, T cell levels slightly decreased early and then recovered 12 h after I/R injury in both the transplantation and control groups with no significant difference observed (Fig. 2A). These data indicated that organoids transplantation gradually exerts a protective effect targeted immune microenvironment after intestinal I/R injury.

Considering that changes in monocytes and macrophages are coordinated, we subsequently assessed monocyte subsets and the function of macrophages present in intestinal tissues 36 h after surgery. Monocytes gradually change their phenotype, reflecting different stages of maturation to tissue-resident macrophages[28–30]. We identified three subsets of monocytes during their recruitment and differentiation into macrophages in the intestinal lamina propria (LP); (Fig. 2B). In the organoids transplantation group, the expression of CX3CR1 and CD206 gradually increased as cells developed from Ly6C+MHCII- inflammatory monocytes into Ly6C-MHCII+ macrophages (Fig. 2C). Consistent with this, the proportion of inflammatory Ly6C+MHCII- monocytes in the transplantation group was significantly lower than that in the control group. Monocyte-derived macrophages, including Ly6C+MHCII+ and Ly6C-MHCII+ cells, in organoid-transplanted mice were significantly increased compared with those in control mice (Fig. 2D). Further analysis of the intestinal macrophage

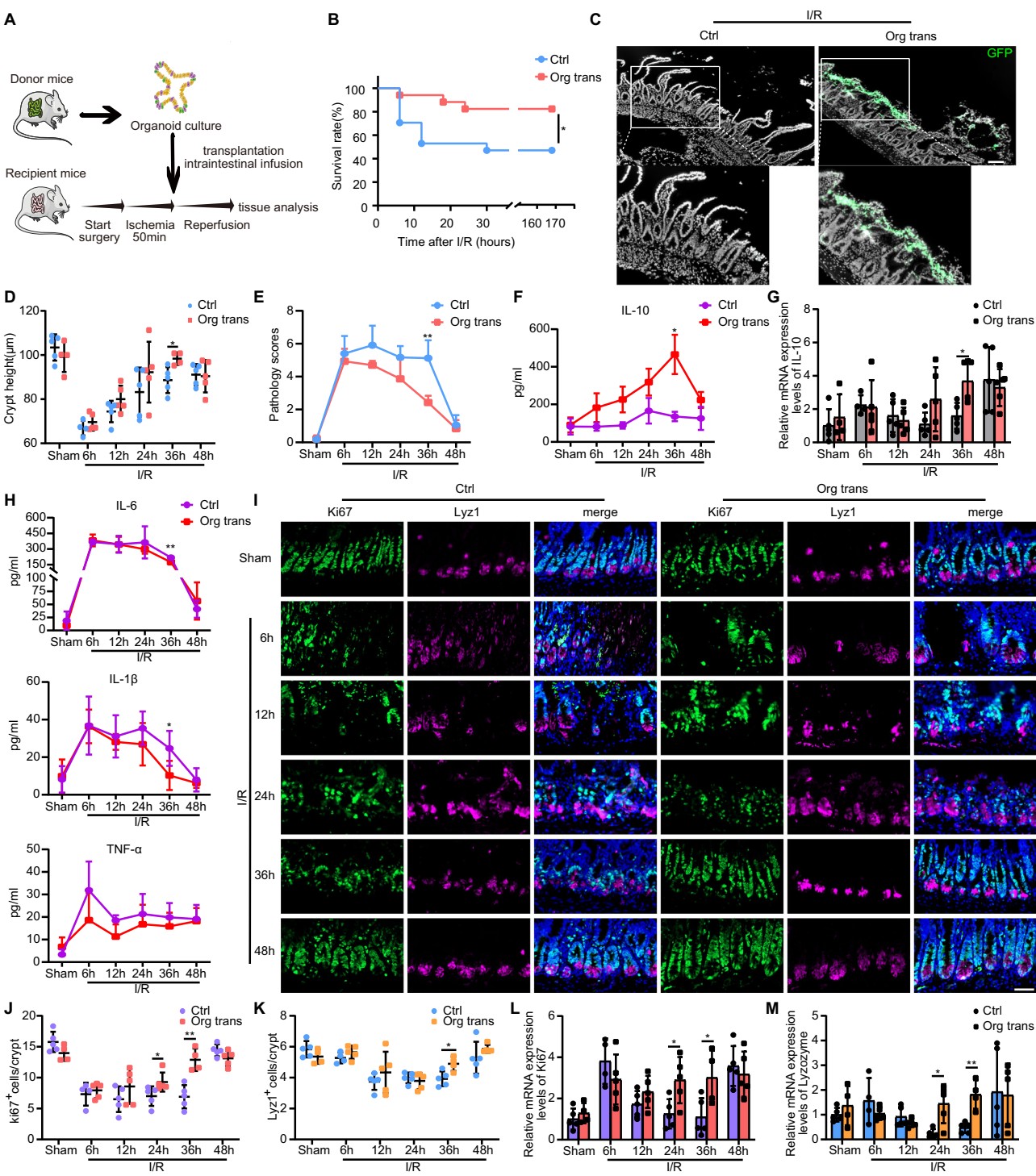

(F4/80+) subset revealed that the relative number and protein expression of CD206+ M2-like macrophages in the organoids transplantation group were significantly increased compared with that in the control group (Fig. 2E−H). Mature macrophages isolated from the LP of transplanted mice also showed increased CD206 (Fig. 2I) and IL-10 (Fig. 2M) mRNA expression compared with those of control mice 36 h after injury. Macrophages in the transplantation group exhibited higher levels of IL-10 compared to the control group, which likely contributed to promoting tissue recovery in the injured intestine (Fig. 2J-L). These findings revealed that organoids transplantation decreased monocyte recruitment and enhanced macrophage polarization toward an M2-like phenotype.

## Organoids transplantation alleviates intestinal I/R injury through macrophages

In order to investigate the functional relationship between tissue macrophages and transplanted organoids, we deleted intestinal macrophages before intestinal I/R modeling by administrating intraperitoneal injections of clodronate liposome thrice (Fig. 3A). Pathological analyses indicated that tissue damage in the phosphate-buffered saline (PBS) liposome-treated transplantation group was improved compared with that in the PBS liposome-treated control group. However, clodronate liposomes treatment reversed the pathological score (Fig. 3B) and inflammatory cytokine levels (Fig. 3C) in the transplantation group compared with that in the PBS liposome-treated

**Fig. 1 | Therapeutic effect of organoid transplantation in intestinal ischemia-reperfusion (I/R). A** Schematic illustration of the induction protocols for intestinal I/R and organoids transplantation. **B** Kaplan-Meier survival curve of organoids-transplanted and control mice after induction of intestinal I/R (n = 17 mice/group). Represent significant *p = 0.0277 using two-tailed log-rank test. **C** Representative images of organoids engraftments in the small intestine 36 h after I/R injury. **D** Statistical analysis of intestinal crypt depth. Five different fields from each mouse were obtained for each group (n = 4 mice for sham group with organoids transplanted, control group at 6 h and transplanted group at 36 h, n = 5 mice for the rest groups). Represent significant *p = 0.0198 using two-tailed student's t test. **E** Quantification of pathology score at indicated time points (n = 4 mice for sham group with organoids transplanted, control group at 6 h and transplanted group at 36 h, n = 5 mice for the rest groups). Represent significant **p = 0.0023 using two-tailed student's t test. Protein concentration (**F**) and mRNA level (**G**) of IL-10 (n = 4 mice for sham group with organoids transplanted, control group at 6 h and transplanted group at 36 h, n = 5 mice for the rest groups). Represent significant *p = 0.0109 using two-tailed student's t test in (**F**); represent significant *p = 0.0199 using two-tailed student's t test in (**G**). **H** Enzyme-linked immunosorbent assay (ELISA) analysis of IL-6, IL-1β and TNF-α protein concentration in serum (n = 4 mice for sham group with organoids transplanted, control group at 6 h and transplanted group at 36 h, n = 5 mice for the rest groups) Represent significant p value using two-tailed student's t test, IL-6: 0.0016; IL-1β: 0.039. **I** Immunofluorescence image of intestinal crypts following Ki-67 and lysozyme staining. Quantification of Ki-67⁺ proliferation cells (**J**) and lyz1⁺ Paneth cells (**K**) (n = 4 mice for sham group with organoids transplanted, control group at 6 h and transplanted group at 36 h, n = 5 mice for the rest groups). Ki-67⁺ proliferation cells: represent significant *p value = 0.0159 using two-tailed Mann–Whitney test; represent significant **p value = 0.0019 using two-tailed student's t test. Lyz1⁺ Paneth cells: represent significant *p value = 0.0181 using two-tailed student's t test. Twenty-five different crypts from each mouse were obtained for each group. Quantitative real-time PCR (qRT-PCR) analysis of intestinal tissue for Ki-67 (**L**) and lysozyme (**M**) mRNA expression (n = 4 mice for sham group with organoids transplanted, control group at 6 h and transplanted group at 36 h, n = 5 mice for the rest groups). Represent significant p value using two-tailed student's from left to right, Ki-67: 0.0249, 0.0365; lysozyme: 0.0103, 0.0019. The statistical tests employed included: two-tailed log-rank test, two-tailed student's t test and Mann–Whitney tests. *P < 0.05, **P < 0.01. Scale bar, 100 μm in (**C**) and 50 μm in (**I**). Each dot represents data from a single mouse ([**D**], [**G**] and [**J**–**M**]). Bar graphs represent mean ± standard deviation (SD). Source data are provided as a Source Data file.

transplantation group. Macrophage depletion failed to produce IL-10 in both clodronate liposome-treated transplantation and control groups (Fig. 3C). Furthermore, clodronate liposome treatment led to a dramatic disruption of the epithelial barrier in the transplantation group and control group (Fig. 3D–F). These results indicated that macrophages are indispensable for the therapeutic efficacy of organoids transplantation in intestinal I/R injury. Moreover, clodronate liposomes administration impaired the ability of transplanted mice to generate Ly6C⁺MHCII⁺ and Ly6C⁻MHCII⁺ macrophages (Fig. 3G, H) and decreased the levels of M2-like macrophages as well (Fig. 3I–K). Collectively, these results indicated that the effects of organoids transplantation rely largely on the LP macrophages.

## L-malic acid (MA) secreted by organoids modulates macrophage polarization

Given the significant treatment efficacy of transplanted organoids during 36 h in intestinal I/R injury, they did not form a new niche and generate villus yet. A previous study has reported that prostate organoids may help maintain male fertility by producing citrate[31]. To investigate whether the organoids employed secretory mechanisms to influence macrophage polarization, we performed experiments with conditioned medium (Fig. 4A). We isolated conditioned medium from organoid cultures and found that the organoid-derived conditioned medium co-culture systems increased IL-10 secretion by bone marrow-derived macrophages (BMDMs) compared with that in the control group (Fig. 4B). Similar changes in CD206 expression were observed by flow cytometry analysis and immunofluorescence of BMDMs (Fig. 4C, D). Furthermore, co-culture of BMDMs with organoid-derived conditioned medium induced upregulation of M2 marker genes, including arginase 1 (*Arg1*), *CD206*, *Ym1/2*, and *IL-10*, whereas no difference was found in genes associated with M1-like phenotypes, such as tumor necrosis factor-alpha (*TNF-α*), *CD86* and *IL-1β*, when compared with co-culture of BMDMs with control medium (Fig. 4E).

We subsequently analyzed the metabolic profile of organoid-derived conditioned medium as well as cecal contents isolated from organoids transplanted mice and control mice. The control medium was used as a background, and the metabolites of the background were subtracted from the organoid-derived conditioned medium samples. A total of 296 metabolites were upregulated under the positive ion mode (Supplementary Data 1) and 193 were upregulated under the negative ion mode (Supplementary Data 2). The highest enrichment in the top 35 metabolites in organoid-derived conditioned medium samples under negative ion mode is shown in Fig. 4F. The variables of different metabolites in mouse cecal contents were

identified as P < 0.05, fold change >1.5, and variable importance in projection (VIP) > 1. Volcano plot analysis demonstrated that 112 metabolites of cecal contents in the transplantation group were significantly upregulated from those in the control group under the negative ion mode (Supplementary Data 3) (Fig. 4G). And 116 metabolites in the transplanted group were significantly upregulated from those in the control group under the positive ion mode (Supplementary Data 4). Sixteen "consensus" metabolites were shared between the organoid-derived conditioned medium and upregulated differential metabolites in the cecal contents of the transplanted group under the positive ion mode (Fig. 4H). Additionally, twelve "consensus" metabolites were shared between the organoid-derived conditioned medium and upregulated differential metabolites in the cecal content of the transplanted group under the negative ion mode (Fig. 4I). Interestingly, the α-ketoglutarate (αKG) and L-malic acid (MA) levels in the transplantation group were elevated relative to those in the control group, suggesting that high rates of oxidation may have flooded the Krebs cycle in transplanted mice.

Indeed, tricarboxylic acid (TCA) cycle metabolites are known to exert antithrombotic, immunomodulatory, and anti-inflammatory effects[32,33]. MA, an intermediate in the TCA cycle, which is recognized as a safe, non-toxic, harmless, and edible organic acid by the U.S. Food and Drug Administration[34], plays an important role in ATP production within mitochondria. Considering the important role played by oxidative phosphorylation in regulating macrophage function[22] and potentially reducing I/R damage[35], we measured target metabolism to verify whether organoids produced MA and αKG. The results indicated that MA levels in the organoid-derived conditioned medium and cecal contents of organoid-transplanted mice were significantly increased compared with those in the control group (Fig. 4J–K). The αKG level in the organoid-derived conditioned medium was significantly increased compared with those in the control group (Supplementary Fig. 3A), while the αKG level in the cecal content did not differ between groups (Supplementary Fig. 3B).

To better understand the clinical correlation between MA and the prognosis of patients undergoing intestinal I/R injury, we collected fecal and blood samples from patients before (T0) and 12 h after (T1) elective cardiac valve replacement or coronary artery bypass graft under cardiopulmonary bypass (CPB) surgery because CPB surgery patients undergoing extracorporeal circulation for several hours tend to develop intestinal ischemia at the time of surgery, or non-occlusive ischemia after surgery, as a result of low cardiac output after surgery[36]. Correlation analysis failed to show a significant correlation between MA content in the feces of patients

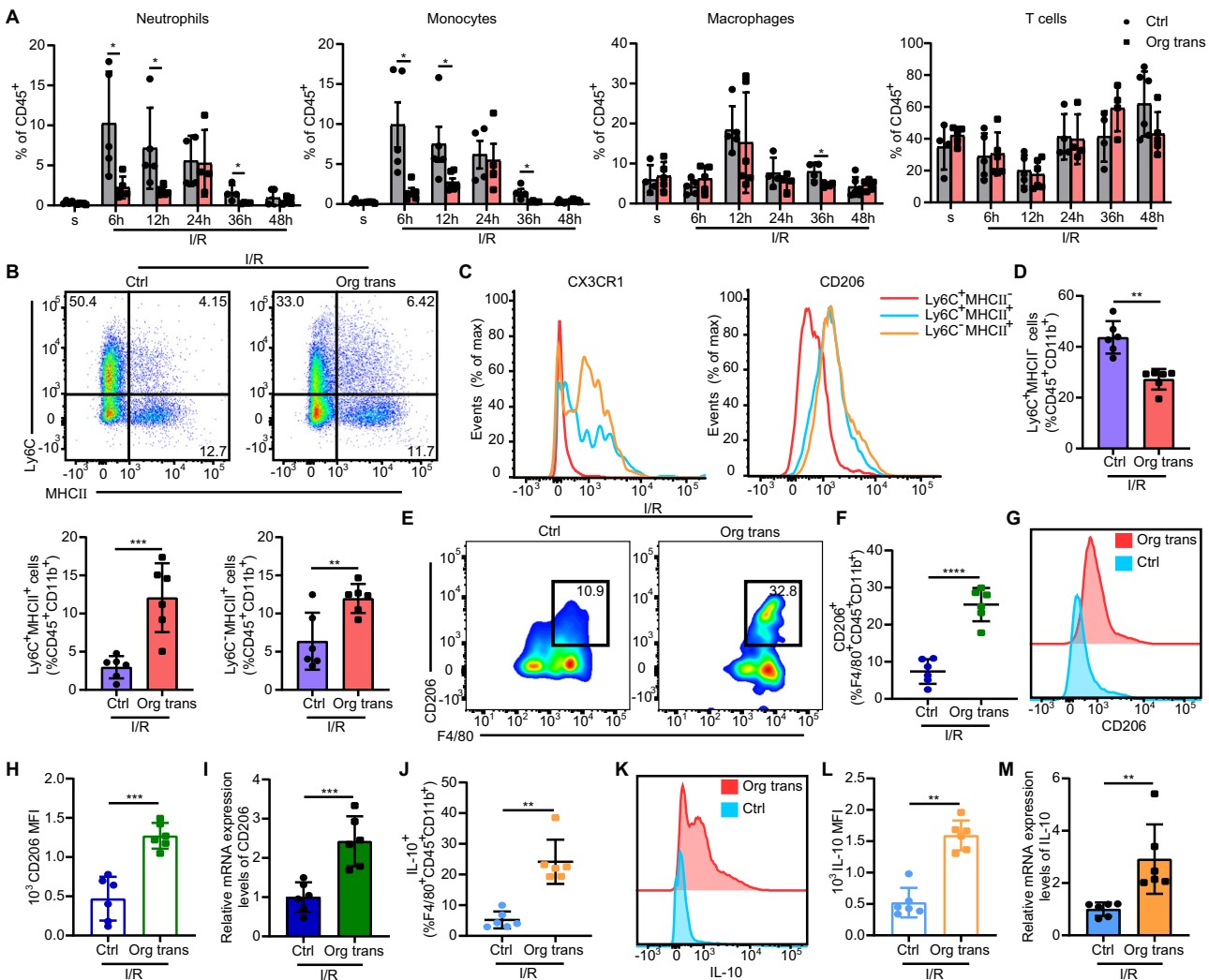

**Fig. 2 | Myeloid landscape is essential for cell therapy. A** Flow cytometry analysis of neutrophils (CD45+CD11b+Ly6G+), monocytes (CD45+CD11b+Ly6C+), macrophages (CD45+CD11b+F4/80+), and T cells (CD45+CD11b-CD3+) in the lamina propria (LP) of the small intestine (n = 5 mice for sham group with organoids transplanted, control group at 6 h, transplanted group at 6 h, control group at 12 h, transplanted group at 48 h, n = 6 mice for transplanted group at 12 h and control group at 48 h, n = 4 mice for the rest groups). Neutrophils: represent significant p value using two-tailed student's from left to right: 0.0270, 0.0293, 0.0406; monocytes: represent significant *p value = 0.0191 using two-tailed student's t test at 6 h, represent significant *p value = 0.0173 using two-tailed Mann–Whitney test at 12 h, represent significant *p value = 0.0345 using two-tailed student's t test at 36 h; macrophages: represent significant *p value = 0.0297 using two-tailed student's t test.
**B** Expression of Ly6C and MHCII by live CD45+CD11b+ small intestinal LP cells in organoid-transplanted and control mice 36 h after intestinal I/R and transplantation. **C** Expression of CX3CR1 and CD206 in the indicated cell subsets from organoid-transplanted mice. **D** Frequency of Ly6C+MHCII-, Ly6C+MHCII+, and Ly6C-MHCII+ subsets among CD45+CD11b+ cells in the small intestine of different groups (n = 6 mice/group). Ly6C+MHCII- cells: represent significant **p value = 0.0022 using two-tailed Mann–Whitney test; Ly6C+MHCII+ cells: represent significant ***p value = 0.0008 using two-tailed student's t test; Ly6C-MHCII+ cells: represent significant **p value = 0.0084 using two-tailed student's t test.

**E** Representative flow cytometric analysis of CD206+ cells in the small intestine LP 36 h after intestinal I/R injury. **F** Quantification of CD206+F4/80+CD45+CD11b+ macrophages (n = 6 mice/group). Represent significant ****p value <0.0001 using two-tailed student's t test. **G** Representative histograms of CD206 expression. **H** Quantification of CD206 mean fluorescence intensity (MFI) in the LP of the small intestine 36 h after intestinal I/R injury (n = 6 mice/group). Represent significant ***p value = 0.0001 using two-tailed student's t test. **I** qRT-PCR analysis of CD206 mRNA in F4/80+ macrophages (n = 6 mice/group). Represent significant ***p value = 0.0008 using two-tailed student's t test. **J** Quantification of IL-10+F4/80+CD45+CD11b+ macrophages (n = 6 mice/group). Represent significant **p value = 0.0022 using two-tailed Mann–Whitney test. **K** Representative histograms of IL-10 expression. **L** Quantification of IL-10 MFI in the small intestine LP 36 h after intestinal I/R injury (n = 6 mice/group). Represent significant **p value = 0.0022 using two-tailed Mann–Whitney test. **M** qRT-PCR analysis of IL-10 mRNA expression in F4/80+ macrophages (n = 6 mice/group). Represent significant **p value = 0.0022 using two-tailed Mann–Whitney test. The statistical tests employed included: two-tailed student's t test and Mann–Whitney tests. *P < 0.05, **P < 0.01, ***P < 0.001, ****P < 0.0001. Each dot represents data from a single mouse ([**A**], [**D**], [**F**], [**H–J**] and [**L**, **M**]). Bar graphs represent mean ± SD. Source data are provided as a Source Data file.

before surgery and the preoperative plasma intestinal fatty-acid binding protein (I-FABP) and D-lactic acid levels, both of which are positive biomarkers of intestinal injury (Supplementary Fig. 3C, D). However, MA content in the preoperative fecal samples was negatively correlated with the Lausanne intestinal failure estimation (LIFE) gastrointestinal injury scores, plasma I-FABP and D-lactic acid concentrations at T1 compared with T0 (Fig. 4L–N), indicating that

reduction in MA was associated with worse gastrointestinal function recovery. We found no significant correlation between the levels of αKG in the preoperative fecal samples with the LIFE gastrointestinal injury scores, and plasma I-FABP at both T0 and T1 compared with that at T0 (Supplementary Fig. 3E-F). Thus, we surmised that MA shows potential as a therapeutic agent against intestinal I/R injury.

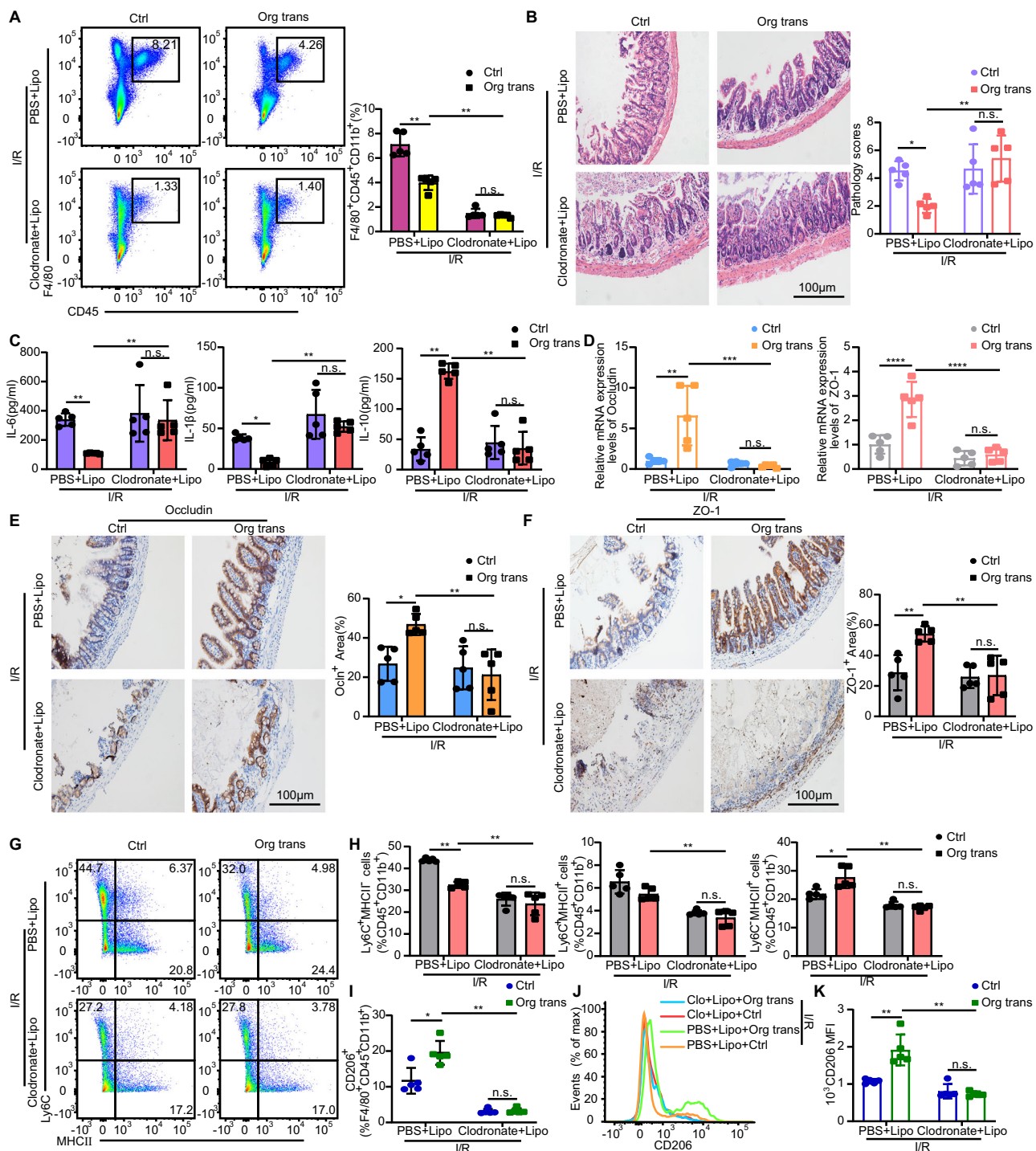

## MA promotes M2 macrophage polarization and enhances mucosal recovery in I/R mice

Next, we investigated whether MA affects intestinal I/R injury. Mice were administrated by gavage at a dose of 250 mg/kg MA per day for five days before intestinal I/R modeling. We found that MA treatment in I/R mice improved pathological damage (Fig. 5A) and barrier tight junction protein and mRNA expression (Fig. 5B–D). Moreover, MA significantly decreased the production of inflammatory cytokines IL-6 and IL-1β compared with that in the vehicle group (Fig. 5E). MA supplementation also restored the expression levels of macrophage CD206 (Fig. 5F–H) and IL-10 protein (Fig. 5I–L). Similar observations were made in vitro, where BMDMs stimulated with MA showed higher

*Arg1*, *Ym1/2*, and *IL-10* expression (Fig. 5M) as well as higher CD206 surface expression, than the vehicle group (Fig. 5N–O). These data indicated that restoring MA levels in the intestine rescues macrophage polarization and enables mucosal recovery.

Furthermore, MA-supplemented mice exhibited a significant increase in Ki-67+ proliferating cells at 36 h compared with that in the vehicle group (Supplementary Fig. 4A). The small intestinal crypts of the MA-supplemented group also exhibited significantly more Paneth, OLFM4+, and goblet cells than those of the vehicle group at 36 h (Supplementary Fig. 4A-C). These data suggested that MA supplementation promotes the self-renewal maintenance of ISCs.

**Fig. 3 | Macrophages are indispensable for cell therapy. A** LP macrophages were analyzed using flow cytometry for F4/80⁺CD45⁺CD11b⁺ cells in mice treated with clodronate liposomes or PBS liposomes 36 h after intestinal I/R injury and transplantation. Representative images and quantification of the macrophage population (n = 5 mice/group). Represent significant *p* value using two-tailed Mann–Whitney test. 0.0079 (PBS + Lipo, Ctrl vs Org trans), 0.0079 (Org trans, PBS + Lipo vs Clodronate + Lipo). **B** Hematoxylin and eosin (H & E) staining of small intestinal tissue from mice 36 h after induction of intestinal I/R and quantification of small intestinal pathology score (n = 5 mice/group). Represent significant *p* value using two-way ANOVA followed by the Tukey test. 0.0326 (PBS + Lipo, Ctrl vs Org trans), 0.0036 (Org trans, PBS + Lipo vs Clodronate + Lipo). **C** ELISA detection of IL-6 and IL-1β, and IL-10 production in mice 36 h after intestinal I/R and transplantation (n = 5 mice/group). Represent significant *p* value using two-tailed Mann–Whitney test for IL-6. 0.0079 (PBS + Lipo, Ctrl vs Org trans), 0.0079 (Org trans, PBS + Lipo vs Clodronate + Lipo). Represent significant *p* value using two-way ANOVA followed by the Tukey test for IL-1β. 0.0334 (PBS + Lipo, Ctrl vs Org trans), 0.0021 (Org trans, PBS + Lipo vs Clodronate + Lipo). Represent significant *p* value using two-tailed Mann–Whitney test for IL-10. 0.0079 (PBS + Lipo, Ctrl vs Org trans), 0.0079 (Org trans, PBS + Lipo vs Clodronate + Lipo). **D** qRT-PCR analysis of Occludin and ZO-1 mRNA in the small intestinal tissue of mice that underwent intestinal I/R and transplantation for 36 h (n = 5 mice/group). Represent significant *p* value using two-way ANOVA followed by the Tukey test for Occludin and ZO-1. For Occludin, 0.0011 (PBS + Lipo, Ctrl vs Org trans), 0.0004 (Org trans, PBS + Lipo vs Clodronate + Lipo). For ZO-1, <0.0001 (PBS + Lipo, Ctrl vs Org trans), < 0.0001 (Org trans, PBS + Lipo vs Clodronate + Lipo). **E** Representative immunohistochemistry images of Occludin staining and quantification of the Occludin staining area (n = 5 mice/group). Represent significant *p* value using two-way ANOVA followed by the Tukey test. 0.0255 (PBS + Lipo, Ctrl vs Org trans), 0.0042 (Org trans, PBS + Lipo vs Clodronate + Lipo). **F** Representative immunohistochemistry images of ZO-1 and quantification of the ZO-1 staining area (n = 5 mice/group). Represent significant *p* value using two-way ANOVA followed by the Tukey test. 0.0040 (PBS + Lipo, Ctrl vs Org trans), 0.0022 (Org trans, PBS + Lipo vs Clodronate + Lipo). **G** Expression of Ly6C and MHCII by live CD45⁺CD11b⁺ small intestinal LP cells in mice 36 h after intestinal I/R and transplantation. **H** Frequency of Ly6C⁺MHCII⁻, Ly6C⁺MHCII⁺, and Ly6C⁻MHCII⁺ subsets among CD45⁺CD11b⁺ cells in the small intestine of different groups (n = 5 mice/group). Represent significant *p* value using two-tailed Mann–Whitney test for Ly6C⁺MHCII⁻, Ly6C⁺MHCII⁺, and Ly6C⁻MHCII⁺ cells. For Ly6C⁺MHCII⁻ cells, 0.0079 (PBS + Lipo, Ctrl vs Org trans), 0.0079 (Org trans, PBS + Lipo vs Clodronate + Lipo). For Ly6C⁺MHCII⁺ cells, 0.0079 (Org trans, PBS + Lipo vs Clodronate + Lipo). For Ly6C⁻MHCII⁺ cells, 0.0159 (PBS + Lipo, Ctrl vs Org trans), 0.0079 (Org trans, PBS + Lipo vs Clodronate + Lipo). **I** Quantification of CD206⁺F4/80⁺CD45⁺CD11b⁺ macrophages (n = 5 mice/group). Represent significant *p* value using two-tailed Mann–Whitney test. 0.0159 (PBS + Lipo, Ctrl vs Org trans), 0.0079 (Org trans, PBS + Lipo vs Clodronate + Lipo). **J** Representative histograms of CD206 expression. **K** Quantification of CD206 MFI in the LP of the small intestine 36 h after intestinal I/R injury (n = 5 mice/group). Represent significant *p* value using two-tailed Mann–Whitney test. 0.0079 (PBS + Lipo, Ctrl vs Org trans), 0.0079 (Org trans, PBS + Lipo vs Clodronate + Lipo). Scale bar, 100 μm. The statistical tests employed included: two-way ANOVA followed by the Tukey test for multiple comparisons and Mann–Whitney tests. *P < 0.05, **P < 0.01, ***P < 0.001, ****P < 0.0001. Each dot represents data from a single mouse ([**A–F**], [**H, I**], and [**K**]). Bar graphs represent mean ± SD. Source data are provided as a Source Data file.

## MA induces M2 polarization in a suppressor of cytokine signaling 2 (SOCS2)-dependent manner

We performed RNA sequencing (RNA-seq) analysis to investigate the transcriptional signatures of BMDMs after administrating MA. We treated BMDMs with 4 μM MA for 24 h and obtained RNA samples for sequencing analysis. Differences in gene expression were mapped using a heatmap. Significant differences were observed between macrophages treated with or without MA. The responses significantly associated with MA-treated BMDMs were involved in arginine metabolism, transforming growth factor beta (TGF-β) receptor binding, and defense responses to gram-negative bacteria (Fig. 6A). In order to investigate the effect of MA on macrophage polarization, we observed suppressor of cytokine signaling (SOCS) family genes. The SOCS gene family exerts important effects on numerous physiological processes, including macrophage polarization[37], and we found that the SOCS genes were significantly upregulated in MA-treated BMDMs compared with those in the vehicle group (Fig. 6B). We validated the differential expression of the SOCS gene family in BMDMs treated with MA and confirmed that the mRNA expression of SOCS2 was consistent with that in the RNA-seq results (Fig. 6C).

To clarify the effects of SOCS2 in vitro, we performed transient transfection of small interfering RNA (siRNA) in BMDM cells. BMDMs treated with siRNA against SOCS2 (siSOCS2) displayed a reduction CD206 expression induced by MA to levels similar to those of the vehicle group (Fig. 6D, E). Moreover, the administration of siSOCS2 impaired the IL-10-producing function of MA-treated BMDMs compared with that in the vehicle group (Fig. 6F–H).

Furthermore, we carried out an experiment in the SOCS2 knocked-out mice to determine whether MA regulates macrophage polarization and intestinal recovery from I/R injury in a SOCS2-dependent manner in vivo. The pathology scores in MA-administrated SOCS2⁻/⁻ mice were significantly higher than that in the MA-administrated WT mice (Fig. 7A). Moreover, SOCS2⁻/⁻ mice developed decreased intestinal barrier integrity (Fig. 7B, C), secreted more IL-6, IL-1β and fewer IL-10 (Fig. 7D). Additionally, SOCS2-deficient mice impaired the ability to generate Ly6C⁺MHCII⁺ and Ly6C⁻MHCII⁺ macrophages (Fig. 7E) and reduced the proportion of M2-like macrophages (Fig. 7F, G). Taken together, these results indicated that MA administration promotes macrophage polarization into M2-like macrophages in a SOCS2 dependent manner.

## Adoptive transfer of MA-administrated WT macrophages reduces intestinal I/R injury in SOCS2⁻/⁻ Mice

To further dissect the function of MA-administrated intestinal macrophages, we conducted an adoptive transfer assay. Macrophages were sorted from MA-administrated WT or SOCS2⁻/⁻ mice undergoing intestinal I/R injury for 36 h. Recipient WT and SOCS2⁻/⁻ mice were administrated with adoptive transfer after the intestinal ischemia (Fig. 7H). We observed less intestinal pathological damage in both WT mice and SOCS2⁻/⁻ mice upon the adoptive transfer of the macrophages from MA-administrated WT mice (Fig. 7I). In line with this result, adoptive transfer of the macrophages from MA-administrated WT mice restored intestinal barrier integrity (Fig. 7J) and IL-10 secretion (Fig. 7K), decreased the serum IL-6 and IL-1β level, whereas adoptive transfer of the macrophages from MA-administrated SOCS2⁻/⁻ mice did not (Fig. 7K). Our data suggest that macrophages from MA-administrated WT mice can functionally contribute to the intestinal I/R injury.

## Discussion

Here, we report that small intestine organoids exploit their unique colonization features to influence the local immune microenvironment and promote the self-renewal of ISCs during intestinal I/R injury. Interestingly, organoids secrete MA as a metabolic substrate, which promotes M2 macrophage polarization and restores IL-10 levels in a SOCS2 dependent manner in intestinal I/R injury (Fig. 8). This is a novel metabolic endocrine mechanism utilized by transplanted organoids in regulating receptor immune microenvironments, which may be applied to treat intestinal I/R injury. In addition, we found that the fecal MA level of patients undergoing CPB surgery was negatively related to the degree of intestinal injury after surgery, which may represent a potential of MA in translation for the treatment of intestinal I/R injury.

Methods of intestinal organoid establishment enable the long-term culture and expansion of different intestinal cells in a three-dimensional structure[11]. To date, both mouse and human intestine-derived organoids have exhibited extraordinary potential in treating

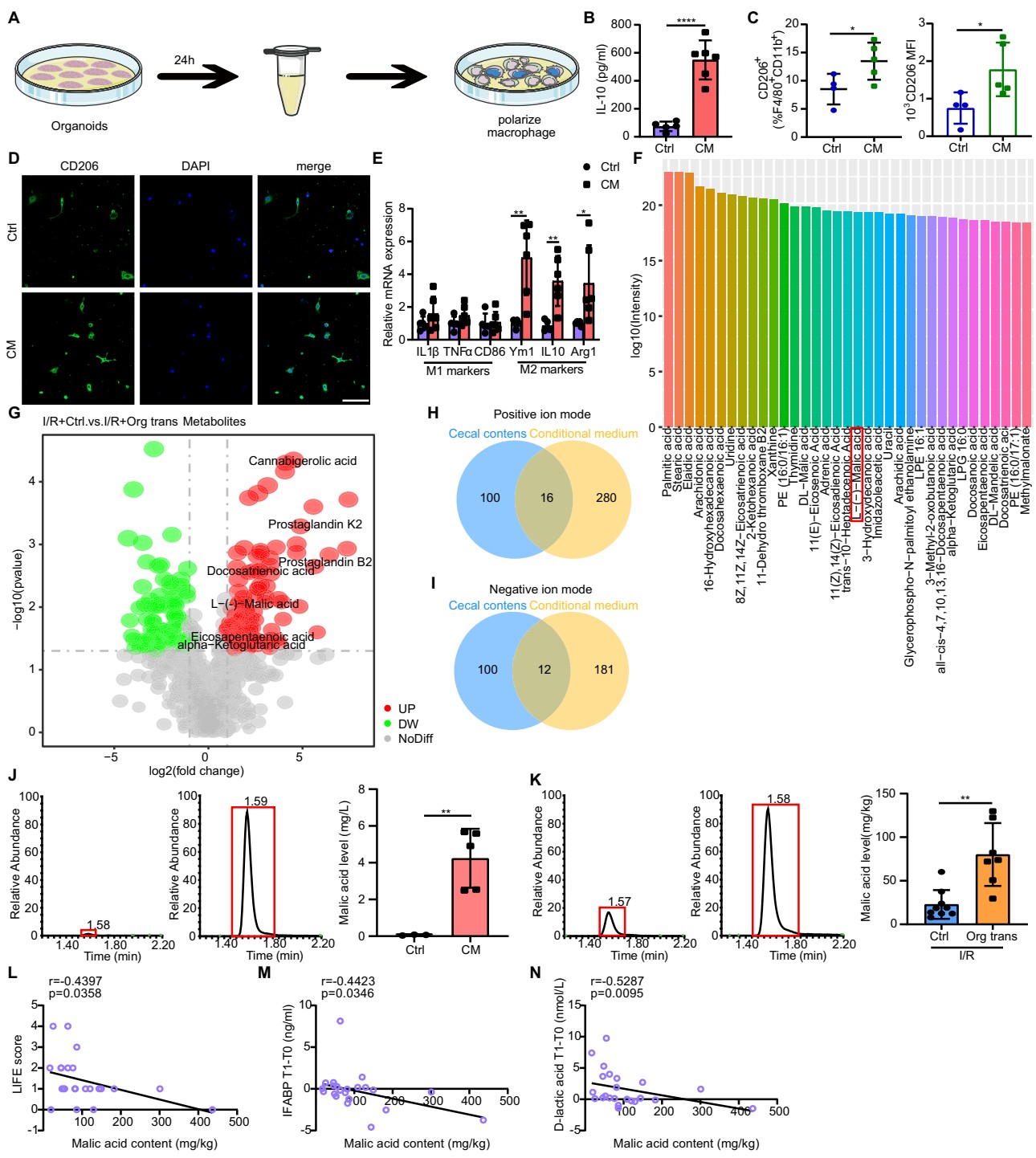

mouse colitis and short bowel syndrome[10,12]. In contrast, transplantation of organoids to resolve small intestinal mucosal injuries has rarely been reported[38]. To conduct a successful transplantation program, we used an inbred mouse model to minimize immunological graft rejection. Mice undergoing intestinal I/R injury slow the movement of the intestinal bowels and enhances transplantation success rates[39]. Previous transplantation experiments suggest that transplanted organoids retain the functional characteristics of their site of origin[10] and that crypt-like domains and stem cell marker expression are not required for successful transplantation[11,40,41]. We chose three alternative grafting materials for transplantation and found that organoids were associated with higher success rates than freshly isolated single Lgr5+ stem cells or crypts, consistent with previous research[11]. We

found that single Lgr5+ stem cells did not relocate to diffusely injured mucosa. This may be because primary ISCs are fragile, leading to a gradual loss of cell viability after the cell sorting process. A single cell may also experience higher hydrodynamic tension and shear forces. Notably, we found that cultured organoids exhibited higher transplantation efficiency and therapeutic effects than freshly isolated crypts. We speculate that this may be because the organoids are cultured for a couple of days and therefore more resistant to the additional manipulation of transplantation, resulting in higher cell viability during the early relocation period[42]. In addition, during the in vitro culture of organoids, the population of stem cells can undergo amplification due to the presence of growth factors and culture conditions that promote their proliferation[43]. This ultimately results in an

**Fig. 4 | Organoid-derived MA controls macrophage polarization. A** Schematic illustration of the experimental set-up. **B** Concentration of IL-10 in the supernatant of BMDMs co-cultured with organoid-derived conditioned or control medium determined by ELISA (n = 5 biological replicates for control group, n = 6 biological replicates for CM group). Represent significant ****p value <0.0001 using two-tailed student's t test. **C** Quantification of CD206⁺F4/80⁺CD11b⁺ macrophages and CD206 MFI (n = 4 biological replicates for control group, n = 5 biological replicates for CM group). CD206⁺F4/80⁺CD11b⁺ macrophages: represent significant *p value = 0.0462 using two-tailed student's t test; CD206 MFI: represent significant *p value = 0.0389 using two-tailed student's t test. **D** Representative images of immunostaining of CD206 (green) and DAPI (blue) in BMDMs stimulated with organoid-derived conditioned or control medium. **E** Genetic profiling of BMDMs stimulated with organoid-derived conditioned medium or control medium (n = 6 biological replicates/group). Represent significant p value using two-tailed student's from left to right: 0.0037, 0.0036, 0.0451. **F** Top 35 metabolites in the organoid-derived conditioned medium group compared with those in the control group under negative ion modes using untargeted liquid chromatography-mass spectrometry (LC-MS) metabolomic analysis. **G** Volcano plot displaying the different metabolites in the cecal contents of organoid-transplanted and control mice under negative ion modes after I/R using untargeted LC-MS metabolomic analysis. Venn diagram displaying the consensus upregulated metabolites in the organoid-derived conditioned medium group and cecal contents of organoid-transplanted mice compared with those in control mice under positive (**H**) and negative (**I**) ion modes, respectively. **J** MA levels in organoid-derived conditioned and control medium groups using targeted LC-MS metabolomic analysis (n = 3 biological replicates for control group, n = 5 biological replicates for CM group). Represent significant **p value = 0.0047 using two-tailed student's t test. **K** MA levels in the cecal contents of the organoid-transplanted and control groups after I/R using targeted LC-MS metabolomic analysis (n = 7 mice for control group, n = 9 mice for transplanted group). Represent significant **p value = 0.0012 using two-tailed Mann–Whitney test. **L** Correlation analysis between preoperative patient fecal MA content and LIFE score (n = 23 samples/group). **M** Correlation analysis between preoperative fecal MA levels and serum I-FABP levels in patients at T1 compared to T0 (n = 23 samples/group). **N** Correlation analysis between preoperative fecal MA levels and serum D-lactate levels in patients at T1 compared to T0 (n = 23 samples/group). Scale bar, 100 μm. The statistical tests employed included: two-tailed student's t test, Spearman's correlation coefficients and Mann–Whitney tests. *P < 0.05, **P < 0.01, ***P < 0.001, ****P < 0.0001. Each dot represents data from a single sample ([**B**, **C**], [**E**], and [**L**–**N**]). Bar graphs represent mean ± SD. Source data are provided as a Source Data file.

increased number of stem cells within the organoid, which can enhance the engraftment rate when transplanted into the recipient.

Here we have developed a postconditioning transplantation method for intestinal I/R injury that is more applicable for clinical transformation. Additionally, we have modified the delivery route of the organoids and optimized the ratio of matrigel to solvent to better fit the intestinal I/R model and ensure the successful transplantation of organoids. This method has the potential to benefit patients through minimally invasive techniques in the future application. Moreover, the standardization protocol for intestinal organoids transplantation and future extensive research will contribute to its application in the treatment of human diseases[44].

Interaction between the recipient immune microenvironment and donor cells plays an important role in the efficacy of disease treatment. For example, retinal organoid-derived progenitor cell transplantation suppresses microglial activation and improves vision during retinal degeneration[27]. Furthermore, M2 macrophages help aged gastric organoids repair gastric ulcers[45]. In the present study, we first demonstrated that the myeloid cell population, rather than T cells, was affected in the transplantation group. Intestinal Ly6Chi monocytes progressively develop into macrophages with the acquisition of MHCII, and loss of Ly6C is accompanied by loss of pro-inflammation and increased IL-10 production[28,46]. Differentiation of monocytes into a subpopulation of macrophages, including MHCIIlo and MHCIIhi macrophages in the gut, occurs more frequently due to their unique environmental niches, which contribute to the removal of dying epithelial cells and tissue remodeling[47,48]. We found that the transplantation group promoted the differentiation of Ly6C⁺MHCII⁻ monocytes into Ly6C⁻MHCII⁺ macrophages. In addition, Ly6C⁻MHCII⁺ cells showed a higher expression of CD206 than Ly6C⁺MHCII⁻ monocytes. Consistent with this result, the transplantation group also showed significantly higher expression levels of CD206 and IL-10 in F4/80⁺ macrophages, which represents an anti-inflammatory state of macrophages, than those in the non-transplanted group. Thus, the differentiation of Ly6C⁺ monocytes to replenish MHCII⁺ macrophages and a higher population of M2-like macrophages reflects their unique role in the immune microenvironment under the influence of transplanted organoids.

The integrity of intestinal barrier structure is closely related to the function of macrophages through various ways[49]. Specifically, the immunoregulatory functions of epithelium, their contribution to the regulation of innate and adaptive immune responses in the intestinal environment to some extent depends on numerous immunoregulatory signals they produced[49,50]. Therefore, we hypothesized that organoids may function via a secretory mechanism influenced the immune microenvironment. Using a conditioned medium, we proved the hypothesis that organoid-derived conditioned medium does promote macrophage polarization and the secretion of IL-10 by macrophages.

Several types of cultured organoids maintain higher metabolism levels than their monolayer cultures[42,51]. This could be attributed to their three-dimensional structure, which promotes increased cell-cell interactions and creates signaling cascades by various cell aggregates[42,52]. Consequently, this enhances the activity of the metabolism. Different metabolites may serve different roles. For example, prostate organoids produce citrate to maintain male fertility[31]. The metabolite αKG, regulates immune responses via epigenetic mechanisms[53]. Furthermore, endogenous gut production was reported to be the major source of TCA cycle metabolites[54]. We found that MA levels in the intestinal organoid-derived conditioned medium, as well as in the cecal content of transplanted mice, were significantly higher compared with those in the controls. The primary source of MA production is predominantly attributed to the oxaloacetate reduction pathway, TCA cycle pathway, and glyoxylate cycle pathway[33,55,56]. We have further shown that MA reduced intestinal damage and promoted the differentiation of Ly6C⁺ monocytes into macrophages. MA also restores IL-10 levels and induces macrophage polarization toward an M2-like phenotype, both in vivo and in vitro. Thus, transplanted organoids shape a pro-stemness and recovery environment to improve epithelial barrier reconstitution through the release of MA. However, multiple functional enzymes, growth factors, and hormones are secreted by mature organoids. Further analyses of the protein profile of organoid-derived conditioned medium may provide more therapeutic options for future applications[23–25]. Macrophage-derived IL-10 subsequently consolidates the homeostasis of the intestinal immune microenvironment by resolving inflammation. Interestingly, deleting or restoring IL-10 levels in organoids-transplanted mice by decreasing or promoting M2-like macrophage polarization (through clodronate liposomes or increasing MA levels), respectively, exerts opposing effects on the mucosal injury. This indicates that the recovery of intestinal injury is highly dependent on increased IL-10 production by anti-inflammatory macrophages.

MA promotes the functional polarization of wound-healing macrophages. M2-like macrophages produce TGF-β, Arg1, and IL-10, which are involved in the resolution of inflammation and tissue regeneration. Arg1⁺ macrophages promote efferocytosis, leading to recovery from inflammation and resolution of atherosclerosis[57]. Arginine metabolism produces polyamines from arginine-1, which are important for cell

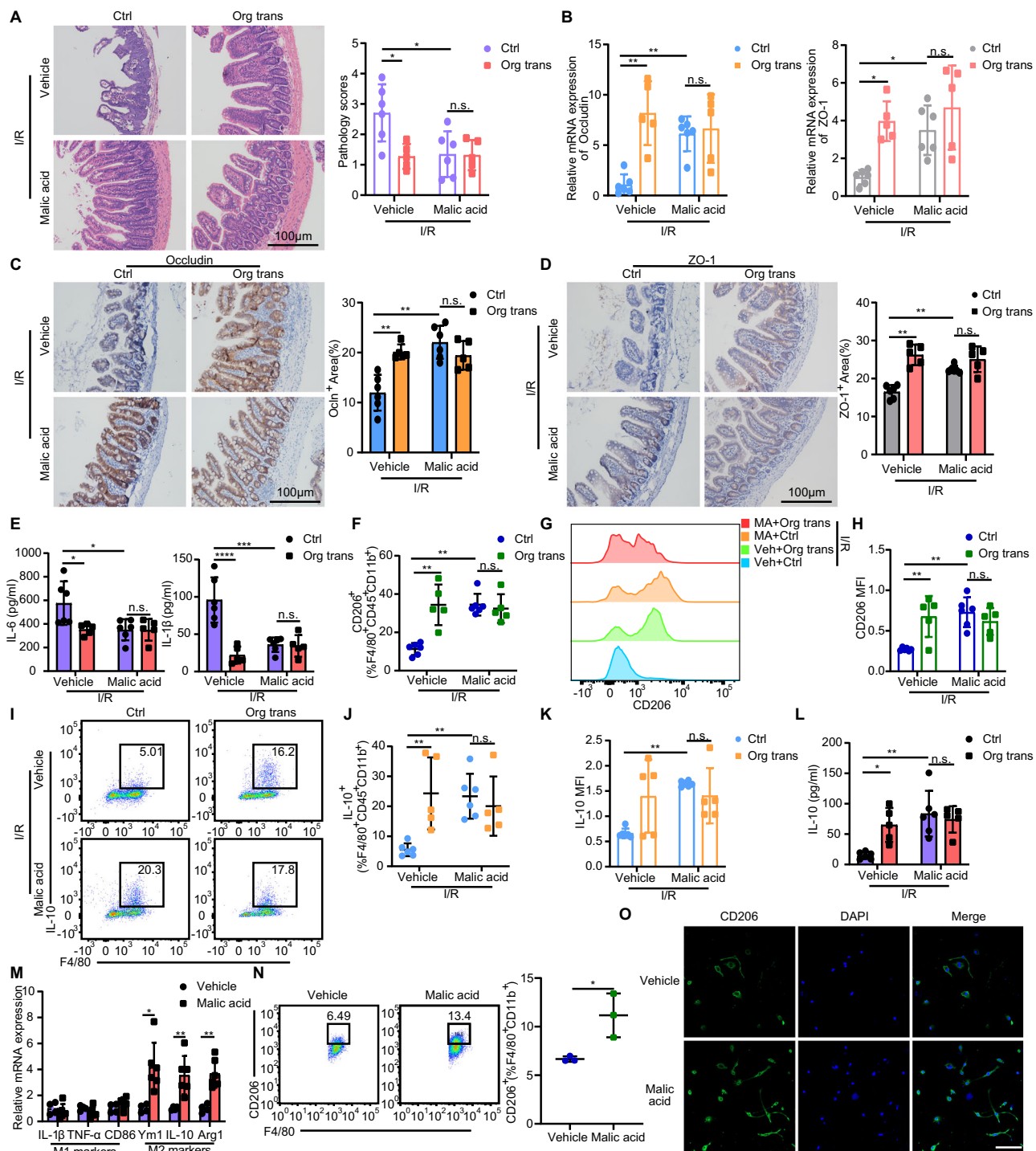

proliferation[57]. TGF-β[+] macrophages enhance muscle regeneration[58] and restore blood barrier function[59]. Upregulation of the IL-1 receptor and its downstream effector's vascular endothelial growth factor (VEGF) and SOCS3 reinforces IL-1 responsiveness and contributes to the anti-inflammatory properties of M2 macrophages[60]. Here, MA significantly increased the expression of Arg1, SOCS2, VEGFα, and TGF-β, suggesting that MA promotes the recovery of injured tissues, which is largely attributed to macrophage M2 polarization. The SOCS family plays an important role in regulating the homeostasis of immune systems due to the preference for the phosphotyrosine motif binding of the SOCS-SH2 domain[61,62]. Although SOCS2 exhibits tremendous potential for suppressing inflammatory diseases[63], its role in intestinal I/R injury has not yet been investigated. Specifically, we report for the

first time that MA regulates macrophage polarization and intestinal recovery from intestinal I/R injury in a SOCS2-dependent manner.

Taken together, our findings suggest that transplanted organoids can alleviate mice intestinal I/R injury by promoting the self-renewal of intestinal stem cells and regulating the immune microenvironment. MA secreted by organoids mediating M2 macrophage polarization in a SOCS2-dependent manner offers new insights into the mechanism underlying the role played by organoids transplantation therapy in treating intestinal I/R injury. This study revealed a novel metabolic endocrine mechanism of transplanted organoids in regulating the immune microenvironment by secreting MA, thereby opening a new avenue for therapeutic strategies to combat intestinal I/R injury in the clinical setting.

**Fig. 5 | Supplementation with MA in I/R mice restores M2 macrophage content and enhances mucosal recovery. A** H & E staining of small intestinal tissue from mice 36 h after induction of intestinal I/R and quantification of the small intestinal pathology score (n = 5 mice for transplanted groups, n = 6 mice for the rest groups). Represent significant *p* value using two-way ANOVA followed by the Tukey test. 0.0167 (Vehicle, Ctrl vs Org trans), 0.0172 (Ctrl, Vehicle vs Malic acid). **B** qRT-PCR analysis of Occludin and ZO-1 mRNA in small intestinal tissues from mice 36 h after intestinal I/R (n = 5 mice for transplanted groups, n = 6 mice for the rest groups). Represent significant *p* value using two-tailed Mann–Whitney test for Occludin. 0.0043 (Vehicle, Ctrl vs Org trans), 0.0043 (Ctrl, Vehicle vs Malic acid). Represent significant *p* value using two-way ANOVA followed by the Tukey test for ZO-1. 0.0105 (Vehicle, Ctrl vs Org trans), 0.0259 (Ctrl, Vehicle vs Malic acid). **C** Representative immunohistochemistry images of Occludin expression and quantification of the area of Occludin immunohistochemical staining (n = 5 mice for transplanted groups, n = 6 mice for the rest groups). Represent significant *p* value using two-tailed Mann–Whitney test. 0.0043 (Vehicle, Ctrl vs Org trans), 0.0022 (Ctrl, Vehicle vs Malic acid). **D** Representative immunohistochemistry images of ZO-1 and quantification of the area of ZO-1 staining (n = 5 mice for transplanted groups, n = 6 mice for the rest groups). Represent significant *p* value using two-tailed Mann–Whitney test. 0.0043 (Vehicle, Ctrl vs Org trans), 0.0022 (Ctrl, Vehicle vs Malic acid). **E** ELISA detection of IL-6 and IL-1β production in mice 36 h after intestinal I/R (n = 5 mice for transplanted groups, n = 6 mice for the rest groups). Represent significant *p* value using two-way ANOVA followed by the Tukey test. For IL-6, 0.0297 (Vehicle, Ctrl vs Org trans), 0.0190 (Ctrl, Vehicle vs Malic acid). For IL-1β, < 0.0001 (Vehicle, Ctrl vs Org trans), 0.0002 (Ctrl, Vehicle vs Malic acid). **F** Quantification of CD206⁺F4/80⁺CD45⁺CD11b⁺ macrophages (n = 5 mice for transplanted groups, n = 6 mice for the rest groups). Represent significant *p* value using two-tailed Mann–Whitney test. 0.0043 (Vehicle, Ctrl vs Org trans), 0.0022 (Ctrl, Vehicle vs Malic acid). **G** Representative histograms of CD206

expression. **H** Quantification of CD206 MFI in the LP of the small intestine 36 h after intestinal I/R injury (n = 5 mice for transplanted groups, n = 6 mice for the rest groups). Represent significant *p* value using two-way ANOVA followed by the Tukey test. 0.0056 (Vehicle, Ctrl vs Org trans), 0.0012 (Ctrl, Vehicle vs Malic acid). **I** Representative histograms of IL-10 expression. **J** Quantification of IL-10⁺F4/80⁺CD45⁺CD11b⁺ macrophages (n = 5 mice for transplanted groups, n = 6 mice for the rest groups). Represent significant *p* value using two-way ANOVA followed by the Tukey test. 0.0083 (Vehicle, Ctrl vs Org trans), 0.0085 (Ctrl, Vehicle vs Malic acid). **K** Quantification of IL-10 MFI in the LP of the small intestine 36 h after intestinal I/R (n = 5 mice for transplanted groups, n = 6 mice for the rest groups). Represent significant *p* value using two-tailed Mann–Whitney test. 0.0022 (Ctrl, Vehicle vs Malic acid). **L** Serum concentration of IL-10 in mice 36 h after intestinal I/R (n = 5 mice for transplanted groups, n = 6 mice for the rest groups). Represent significant *p* value using two-way ANOVA followed by the Tukey test. 0.0247 (Vehicle, Ctrl vs Org trans), 0.0013 (Ctrl, Vehicle vs Malic acid). **M** Genetic profiling of BMDMs stimulated with or without MA (n = 4 biological replicates for vehicle group, n = 6 biological replicates for MA group). Represent significant *p* value using two-tailed student's from left to right: 0.0108, 0.0096, 0.0015. **N** Representative images of CD206 expression and quantification of CD206⁺F4/80⁺CD11b⁺ in BMDMs stimulated with or without MA supplementation (n = 3 biological replicates/group). Represent significant **p* value = 0.0266 using two-tailed student's *t* test. **O** Representative images of immunostaining of CD206 (green) and DAPI (blue) in BMDMs stimulated with or without MA supplementation. Scale bar, 100 μm. The statistical tests employed included: two-way ANOVA followed by the Tukey test for multiple comparisons, two-tailed student's *t* test and Mann–Whitney tests. **P* < 0.05, ***P* < 0.01, ****P* < 0.001, *****P* < 0.0001. Each dot represents data from a single sample ([**A–F**], [**H**], and [**J–N**]). Bar graphs represent mean ± SD. Source data are provided as a Source Data file.

## Methods

Mice with the same genotype were assigned randomly to each group by an independent experimenter, but no specific randomization method was employed. Additionally, no statistical methods were utilized to pre-estimate the sample size. Experimenters responsible for histological analyses, IF and relevant laboratory tests were blinded to the grouping of the mice.

### Animals

Male C57BL/6 J WT mice (6-8 weeks old) were provided by the Animal Center at the Nanfang Hospital of Southern Medical University (Guangzhou, China). Lgr5-EGFP-IRES-CreERT2 (Lgr5-GFP) mice were obtained from Jackson Laboratory (Ban Harbor, ME, USA). Breeding pairs of SOCS2 knocked-out mice were kindly provided by Zai-Long Chi (Wenzhou Medical University). SOCS2⁻/⁻ mice in the C57BL/6 genetic background were generated by using the CRISPR/Cas9-mediated genome engineering technology. The deletion of the exon 3 fragment was carried out using gRNA1 (TTG GCA GTC GTT TTT CTA GT CGG) and gRNA2 (ATT CAG CTA AAA CTA CCT AA GGG) generated by Cyagen Biosciences (Guangzhou, China). All animals were housed in an animal room under specific pathogen-free (SPF) conditions (12 h light-dark cycle, temperature: 22 ± 1 °C and humidity: 55 ± 5%) and had ad libitum access to standard mouse chow (MD17121, MEDICIENCE, Jiangsu, China) and water. The animals used for the experiment were assigned randomly to the different groups. All animal procedures described here were approved by the local Animal Care and Use Committee of the Nanfang Hospital of Southern Medical University (Guangzhou, China) and carried out in line with the National Institutes of Health guidelines. All animal handling were according to the policy of the China Animal Welfare guidelines and under the supervision of the ethics committee of Southern Medical University. Mice were gavaged with 0.2 mL/20 g ddH₂O or 250 mg/kg MA (M7397, Sigma-Aldrich, St. Louis, MO, USA) once a day consecutively for five days until surgery. The mice used here were randomized into the following experimental groups: (i) control; (ii) organoid transplantation; (iii) PBS

liposome; (iv) clodronate liposome; (v) vehicle; and (vi) MA. Mice in the vehicle groups were gavaged 0.2 mL/20 g of ddH₂O once a day for five days and used as the vehicle. Approval of this study was obtained from the Nanfang Hospital animal ethics committee (approval number: IACUC-LAC-20220508-001).

### Establishment of the mouse intestinal I/R injury model

Before surgery, all animals were deprived of food for 18 h, but with ad libitum access to water. Mice were anesthetized using 2-3% inhalant isoflurane, and their skin was prepared before surgery. To establish the model, the small intestine was exposed through a mid-abdominal incision, and the superior mesenteric artery (SMA) was clipped for 50 min using a non-invasive microvascular artery clamp, followed by reperfusion, as previously described[4]. In brief, successful ischemia was identified through the pale color of the small intestine mucosa, while valid reperfusion was determined based on the reappearance of pink color. For liquid resuscitation, 1 mL of saline was immediately injected subcutaneously after the release of the clamp. After the abdominal incision was closed with suture, 1 mg/ml butorphanol (Jiangsu Hengrui Medicine Co., Ltd., Jiangsu, China) was injected subcutaneously for pain management. The survival of the mice was monitored every 6 hours after releasing the artery clamp and continued to be monitored for a study period of 7 days.

### Culture of mouse intestinal organoids

Mice were sacrificed, the small intestine excised, immersed in sterile cold PBS, cut lengthwise, and rinsed using sterile cold PBS to wash away the intestinal contents. The intestine was then cut into pieces (2-4 mm) and placed in a 50-mL centrifuge tube which contained sterile cold PBS (prepared in advance), and a 2-ml pipette was used to wash the pieces via pipetting up and down repeatedly 15-20 times with sterile cold PBS. Then pieces were incubated for 15 min with 10 mL of 30 mM EDTA in PBS at room temperature. Subsequently, the reagent was removed, the intestinal sections were washed with PBS, and the supernatant fractions enriched in crypts were collected using a 70-μm

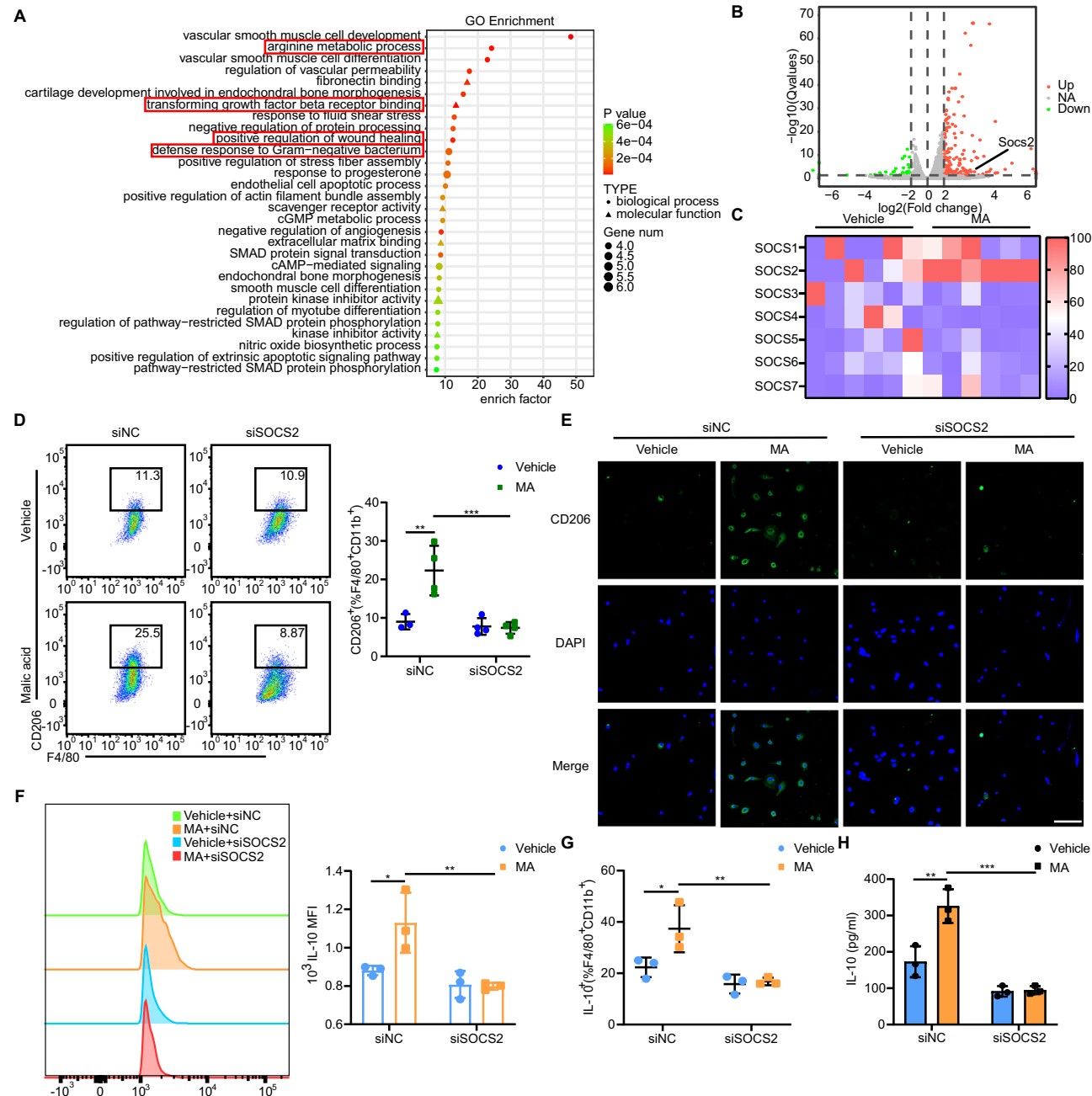

**Fig. 6 | MA-induced macrophage polarization via SOCS2 in vitro. A** Gene ontology (GO) pathway enrichment analyses of the upregulated DEGs between two groups. The size of the node represents the number of DEGs, and the color of the node represents the corresponding P value. **B** Scatter plots comparing the DEGs between the MA administrated and vehicle groups. Genes were monitored according to their expression levels (log10 intensity). Red and green dots represented upregulated and downregulated genes, respectively. **C** Genetic profiling of SOCS family genes in BMDMs administrated with or without MA. **D** Representative histograms and quantification of CD206 expression in BMDMs depleted of SOCS2 using siRNA and co-cultured with or without MA (n = 3 biological replicates for siNC administrated vehicle group, n = 4 biological replicates for the rest groups). Represent significant p value using two-way ANOVA followed by the Tukey test. 0.0034 (siNC, Vehicle vs MA), 0.0007 (MA, siNC vs siSOCS2). **E** Representative images of immunostaining of CD206 (green) and DAPI (blue) in BMDMs depleted of SOCS2 using siRNA and co-cultured with or without MA. **F** Representative histograms of IL-

10 expression and quantification of IL-10+ MFI (n = 3 biological replicates/group). Represent significant p value using two-way ANOVA followed by the Tukey test. 0.0335 (siNC, Vehicle vs MA), 0.0072 (MA, siNC vs siSOCS2). **G** Quantification of IL-10+F4/80+CD45+CD11b+ macrophages in BMDMs depleted of SOCS2 using siRNA and co-cultured with or without MA (n = 3 biological replicates/group). Represent significant p value using two-way ANOVA followed by the Tukey test. 0.0353 (siNC, Vehicle vs MA), 0.0066 (MA, siNC vs siSOCS2). **H** ELISA detection of IL-10 concentration released by BMDMs depleted of SOCS2 using siRNA and co-cultured with or without MA (n = 3 biological replicates/group). Represent significant p value using two-way ANOVA followed by the Tukey test. 0.0020 (siNC, Vehicle vs MA), 0.0001 (MA, siNC vs siSOCS2). Scale bar, 100 μm. The statistical tests employed included: two-way ANOVA followed by the Tukey test for multiple comparisons and two-tailed student's t test. *P < 0.05, **P < 0.01, ***P < 0.001. Each dot represents data from a single sample ([**D**], [**F**–**H**]). Bar graphs represent mean ± SD. Source data are provided as a Source Data file.

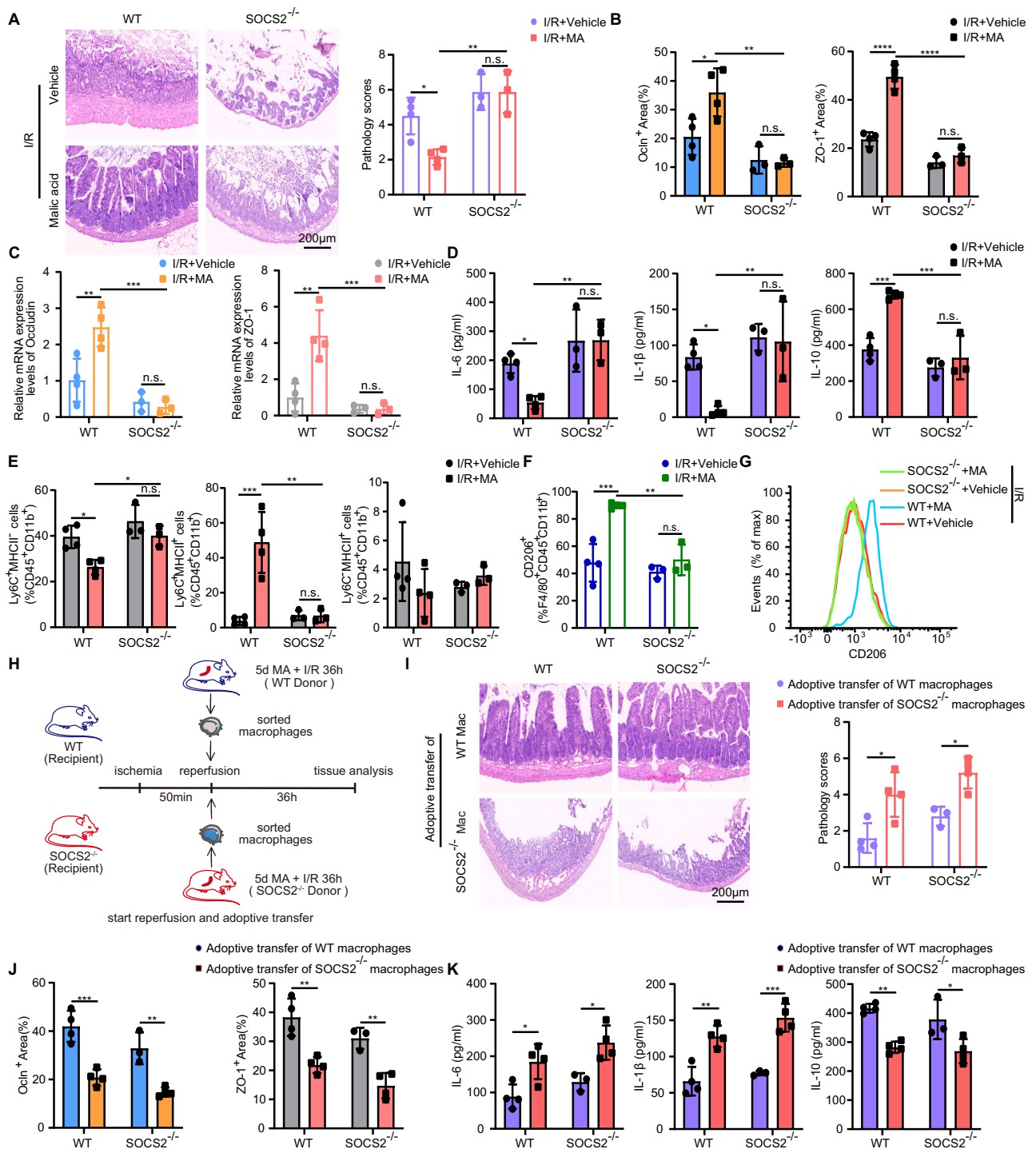

cell strainer. These fractions were centrifuged at $400 \times g$ for 5 min, following which the supernatant was removed. The precipitate was resuspended in 5 mL cold Dulbecco's Modified Eagle Medium/Nutrient Mixture F-12 (DMEM/F12) solution (Gibco, Thermo Fisher Scientific, Waltham, MA, USA), crypts were counted using an inverted microscope, and the required amount of liquid was aspirated into a centrifuge tube and centrifuged for 5 min at $400 \times g$. The precipitate was collected and resuspended in the same amount of organoid medium, including mouse epidermal growth factor (mEGF), Noggin, and R-spondin (ENR medium), and Matrigel (356231, BD Biosciences, Franklin Lakes, NJ, USA). The ENR medium contained DMEM/F12 media (Gibco), Primocin (100 µg/mL, InvivoGen, San Diego, CA, USA), N2 supplement (1X, Gibco), B27 supplement (1X, Gibco), mouse

epidermal growth factor (50 ng/mL, PeproTech, Cranbury, NJ, USA), R-Spondin 1 (500 ng/mL, R&D Systems, Minneapolis, MN, USA), and murine Noggin (100 ng/mL, R&D Systems). Next, the suspension (50 µL) was quickly inoculated on preheated cell culture dishes to form a dome-shaped gelatinous structure. The dishes were then placed in a 37 °C incubator for 20 min. ENR medium was added to the culture dishes for organoid culture after the Matrigel was solidified.

## Orthotopic transplantation

On transplantation day, freshly isolated crypts and sorted small-intestinal Lgr5+ stem cells were prepared as described in the methods. Mature mouse organoids were removed from the Matrigel and washed with cold DMEM/F12 solution. The isolated organoids, freshly isolated

**Fig. 7 | MA alleviates intestinal I/R injury by regulating macrophage polarization in a SOCS2-dependent manner. A** H & E staining of small intestinal tissue from mice 36 h after induction of intestinal I/R and quantification of the small intestinal pathology score (n = 3 in SOCS2$^{-/-}$ mice/group, n = 4 in WT mice/group). Represent significant $p$ value using two-way ANOVA followed by the Tukey test. 0.0241 (WT, I/R + Vehicle vs I/R + MA), 0.0020 (I/R + MA, WT vs SOCS2$^{-/-}$). **B** Quantification of the area of Occludin, ZO-1 immunohistochemical staining (n = 3 in SOCS2$^{-/-}$ mice/group, n = 4 in WT mice/group). Represent significant $p$ value using two-way ANOVA followed by the Tukey test. For Occludin, 0.0231 (WT, I/R + Vehicle vs I/R + MA), 0.0019 (I/R + MA, WT vs SOCS2$^{-/-}$); for ZO-1, < 0.0001 (WT, I/R + Vehicle vs I/R + MA), < 0.0001 (I/R + MA, WT vs SOCS2$^{-/-}$). **C** qRT-PCR analysis of Occludin and ZO-1 mRNA in small intestinal tissues from mice 36 h after intestinal I/R (n = 3 in SOCS2$^{-/-}$ mice/group, n = 4 in WT mice/group). Represent significant $p$ value using two-way ANOVA followed by the Tukey test. For Occludin, 0.0056 (WT, I/R + Vehicle vs I/R + MA), 0.0005 (I/R + MA, WT vs SOCS2$^{-/-}$); for ZO-1, 0.0014 (WT, I/R + Vehicle vs I/R + MA), 0.0007 (I/R + MA, WT vs SOCS2$^{-/-}$). **D** ELISA detection of IL-6, IL-1β and IL-10 production in mice 36 h after intestinal I/R (n = 3 in SOCS2$^{-/-}$ mice/group, n = 4 in WT mice/group). Represent significant $p$ value using two-way ANOVA followed by the Tukey test. For IL-6, 0.046 (WT, I/R + Vehicle vs I/R + MA), 0.0046 (I/R + MA, WT vs SOCS2$^{-/-}$); for IL-1β, 0.0159 (WT, I/R + Vehicle vs I/R + MA), 0.0052 (I/R + MA, WT vs SOCS2$^{-/-}$); for IL-10, 0.0005 (WT, I/R + Vehicle vs I/R + MA), 0.0003 (I/R + MA, WT vs SOCS2$^{-/-}$). **E** Frequency of Ly6C$^+$MHCII$^-$, Ly6C$^+$MHCII$^+$, and Ly6C$^-$MHCII$^+$ subsets among CD45$^+$CD11b$^+$ cells in the small intestine of different groups (n = 3 in SOCS2$^{-/-}$ mice/group, n = 4 in WT mice/group). Represent significant $p$ value using two-way ANOVA followed by the Tukey test for Ly6C$^+$MHCII$^-$ and Ly6C$^+$MHCII$^+$cells. For Ly6C$^+$MHCII$^-$ cells, 0.0169 (WT, I/R + Vehicle vs I/R + MA), 0.0221 (I/R + MA, WT vs SOCS2$^{-/-}$); for Ly6C$^+$MHCII$^+$cells, 0.0004 (WT, I/R + Vehicle vs I/R + MA), 0.0011 (I/R + MA, WT vs SOCS2$^{-/-}$). **F** Quantification of CD206$^+$F4/80$^+$CD45$^+$CD11b$^+$ macrophages (n = 3 in SOCS2$^{-/-}$ mice/group, n = 4 in WT mice/group). Represent significant $p$ value using two-way ANOVA followed by the Tukey test. 0.0004 (WT, I/R + Vehicle vs I/R + MA), 0.0012 (I/R + MA, WT vs SOCS2$^{-/-}$). **G** Representative histograms of CD206 expression in the LP of the small intestine 36 h after intestinal I/R injury. **H** Schematic illustration of the induction protocols for macrophage adoptive transfer. **I** H & E staining of small intestinal tissue from mice 36 h after induction of macrophages adoptive transfer and quantification of the small intestinal pathology score (n = 3 for recipient SOCS2$^{-/-}$ mice adoptive transfer of the macrophages from MA-administrated WT mice group, n = 4 mice for the rest groups). Represent significant $p$ value using two-way ANOVA followed by the Tukey test. 0.0163 (WT, Adoptive transfer of WT macrophages vs Adoptive transfer of SOCS2$^{-/-}$ macrophages), 0.0258 (SOCS2$^{-/-}$, Adoptive transfer of WT macrophages vs Adoptive transfer of SOCS2$^{-/-}$ macrophages). **J** Quantification of the area of Occludin, ZO-1 immunohistochemical staining (n = 3 for recipient SOCS2$^{-/-}$ mice adoptive transfer of the macrophages from MA-administrated WT mice group, n = 4 mice for the rest groups). Represent significant $p$ value using two-way ANOVA followed by the Tukey test. For Occludin, 0.0003 (WT, Adoptive transfer of WT macrophages vs Adoptive transfer of SOCS2$^{-/-}$ macrophages), 0.0023 (SOCS2$^{-/-}$, Adoptive transfer of WT macrophages vs Adoptive transfer of SOCS2$^{-/-}$ macrophages); for ZO-1, 0.0018 (WT, Adoptive transfer of WT macrophages vs Adoptive transfer of SOCS2$^{-/-}$ macrophages), 0.0033 (SOCS2$^{-/-}$, Adoptive transfer of WT macrophages vs Adoptive transfer of SOCS2$^{-/-}$ macrophages). **K** ELISA detection of IL-6, IL-1β and IL-10 production in mice 36 h after macrophages adoptive transfer (n = 3 for recipient SOCS2$^{-/-}$ mice adoptive transfer of the macrophages from MA-administrated WT mice group, n = 4 mice for the rest groups). Represent significant $p$ value using two-way ANOVA followed by the Tukey test. For IL-6, 0.0288 (WT, Adoptive transfer of WT macrophages vs Adoptive transfer of SOCS2$^{-/-}$ macrophages), 0.0226 (SOCS2$^{-/-}$, Adoptive transfer of WT macrophages vs Adoptive transfer of SOCS2$^{-/-}$ macrophages); for IL-1β, 0.0012 (WT, Adoptive transfer of WT macrophages vs Adoptive transfer of SOCS2$^{-/-}$ macrophages), 0.0004 (SOCS2$^{-/-}$, Adoptive transfer of WT macrophages vs Adoptive transfer of SOCS2$^{-/-}$ macrophages); for IL-10, 0.0022 (WT, Adoptive transfer of WT macrophages vs Adoptive transfer of SOCS2$^{-/-}$ macrophages), 0.0154 (SOCS2$^{-/-}$, Adoptive transfer of WT macrophages vs Adoptive transfer of SOCS2$^{-/-}$ macrophages). Scale bar, 200 μm. The statistical tests employed included: two-way ANOVA followed by the Tukey test for multiple comparisons. *$P$ < 0.05, ** $P$ < 0.01, ***$P$ < 0.001, ****$P$ < 0.0001. Each dot represents data from a single sample ([A–F], [I–K]). Bar graphs represent mean ± SD. Source data are provided as a Source Data file.

crypts and sorted small-intestinal Lgr5$^+$ stem cells were then resuspended in Matrigel and DMEM/F12 (1:4) and kept on ice for no more than 30 min. Approximately 3×10$^6$ sorted small intestinal Lgr5$^+$ stem cells, 8000 freshly isolated crypts, and 5000 organoids per 200 μL of fluid were obtained. Control DMEM/F12 and Matrigel were prepared under the same conditions but in the absence of organoids. Mice were anesthetized prior to induction of intestinal I/R injury. Immediately after the clips were released, 200 μL organoid solution was injected once into the duodenal lumen from the proximal to the distal portion of the small intestine using a 1-mL syringe with a 23-gauge needle and the control DMEM/F12 and Matrigel without organoids were injected in the same manner. Subsequently, the small intestine was carefully returned to the abdominal cavity, the incision was closed, and the anus was attached using tissue glue. Six hours after surgery, the glue was removed.

### Blood sample and tissue processing
The mice were anesthetized before being sacrificed. Collection of blood samples was accomplished by removing the eyeballs, following which the blood samples were centrifuged for 15 min at 400 × $g$ to collect sera. After the small intestines were removed, intestinal tissue was fixed using paraformaldehyde (PFA) for histopathological analysis. The lumen was then flushed with cold PBS, and the remaining tissue was collected and frozen using liquid nitrogen. After the cecum was excised, its contents were carefully squeezed out and harvested using sterile forceps. All collected samples were stored at −80 °C before use.

### *Fluorescence* imaging
Prior to fluorescence detection, the entire small intestine was dissected from the mice. Fluorescence was detected using an Ami HTX Optical Imaging System (Spectral Instruments Imaging, Tucson, AZ, USA), and images were taken at 470 nm excitation, 570 nm emission, and 60 s exposure time using small binning (resolution). All images were analyzed and processed using ANALYSIS ONLY Aura 4.0.7 M.

### Organoid-derived conditioned medium
Mouse intestinal organoids were cultured in six-well plates expanded for five to seven days. Then the organoids were cultured in the medium refreshed by DMEM/F12 in advance for 24 h, and the medium was collected and filtered through a 0.22 μm filter (431219, CORNING, Corning, NY, USA) after the 24 h culture to obtain the organoid-conditioned medium.

### Histology and immunohistochemistry
For frozen-embedded sections, 4% PFA (Sigma-Aldrich) was used for the fixation of tissue and organoid samples, and 15% and 30% sucrose solutions were used to sequentially dehydrate samples, which were then embedded, frozen in OCT (Sakura Finetek), and sectioned at 6 μm intervals. For paraffin-embedded sections, 4% PFA was used to fix the samples, after that, samples were dehydrated using an ascending alcohol gradient, embedded in paraffin, and sectioned into 4 μm slices. Sample sections were then stained with hematoxylin and eosin (H & E). Evaluation of the degree of small intestinal injury was accomplished using Chiu's method with some modifications[64]. In brief, the modification pathological scoring criteria are as follows: 0, normal intestinal mucosa and intestinal villus. 1, formation of Gruenhagen cavity begins at the tip of the intestinal villus. 2, formation of Gruenhagen cavity and slight damage of glands. 3, formation and enlargement of subepithelial gaps, congested and engorged capillaries. 4, moderate detachment of epithelium from lamina propria and gland damage. 5, partial loss of villus at the tip. 6, obvious loss of villus and dilated capillaries. 7, loss of lamina propria and significant gland damage. 8, beginning of digestion and decomposition of the lamina propria. 9, hemorrhage and formation of ulcers. The depth of the crypt was

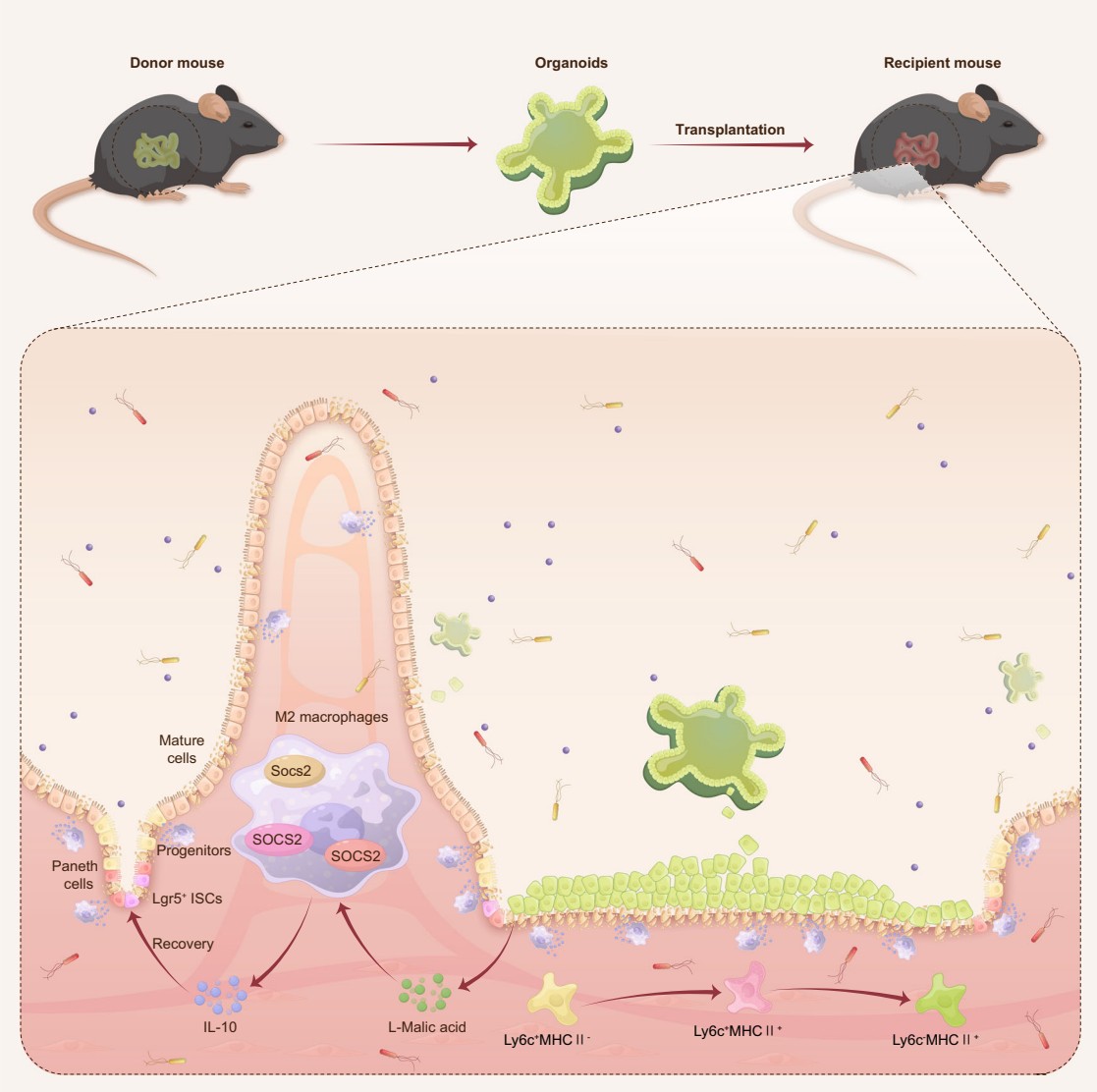

**Fig. 8 | Therapeutic effects of intestinal organoid transplantation.** Intestinal organoids transplanted in small intestinal I/R mucosal injury influence the recipient's local immune microenvironment and self-renewal of ISCs by secreting MA to promote M2 polarization in a SOCS2-dependent manner. Organoids transplantation provides a promising therapeutic strategy for small intestinal mucosal injury.

measured from the bottom to the opening. Paraffin-embedded sections were pressure-cooked in Tris-EDTA (pH 9.0) for 10 min before use. For permeabilization, cells and sections were incubated using 0.1% or 0.3% Triton X-100 (Thermo Fisher Scientific) in PBS at room temperature for 10 min and then washed and blocked in 10% goat serum (Gibco) at room temperature for 1 h. The primary and secondary antibodies used were diluted in 3% bovine serum albumin (BSA) in PBS. The primary antibodies that were used are as follows: anti-ZO-1 (1:400, ab216880, Abcam), anti-occludin (1:200, ab216327, Abcam), anti-Ki-67 (1:200, ab279653, Abcam), anti-OLFM4 (1:200, 39141 S, Cell Signaling Technology), anti-Muc2 (1:100, 27675-1-AP, Proteintech), anti-lysozyme (1:500, A0099, Dako) and anti-mannose receptor (1:200, ab64693, Abcam); DAPI (1:500, D9542, Sigma-Aldrich) was used to stain the nuclei. The positive cells were quantified in five randomly chosen crypts per image. Images were captured using an ortho fluorescence microscope (Axio Imager D2, Carl Zeiss), analyzed, and processed via ImageJ software (National Institutes of Health, Bethesda, MD, USA).

### RNA extraction and quantitative real-time polymerase chain reaction (qRT-PCR)

TRIzol reagent (10296028; Thermo Fisher Scientific) was used to extract the total RNA from the tissues, organoids, and cells. RNA purity and concentration were measured using a spectrophotometer (Nano Drop OneC; Thermo Fisher Scientific). Reverse transcription of RNA to complementary DNA (cDNA) was performed using a cDNA reverse transcription kit (FSQ-101, TOYOBO). Real-time PCR was performed using an ABI Q6 Real-Time PCR System (Applied Biosystems, Foster City, CA, USA), according to the SYBR Green detection protocol (QPK-201, TOYOBO). Finally, 18 S ribosomal RNA was used as a housekeeping gene, and the $2^{-\Delta\Delta CT}$ method was used to normalize data. Primers used here are listed (Supplementary Table 1).

### Cytokine measurements

Cell culture medium and serum samples were thawed at room temperature. Next, the concentrations of IL-6, IL-1β, TNF-α, and IL-10 in the samples were measured using enzyme-linked immunosorbent assay

(ELISA) kits (KE10002, KE10003, KE10007, and KE10008, Proteintech) in accordance with the manufacturer's instructions.

## Flow cytometry

The small intestine was washed using PBS, and Peyer's patches and fat tissue were removed[47]. The intestines were opened lengthwise, cut into pieces and incubated with 10 mL, 5 mM EDTA for 15 min, and the supernatant containing epithelial cells was discarded. The remaining tissues were transferred into RPMI-1640 medium containing collagenase IV (0.5 mg/mL, C5138, Sigma-Aldrich), DNase I (100 U, 10104159001, Roche), and incubated at 37 °C for 1 h. Subsequently, the supernatants were discarded, the tissue fragments were washed repeatedly with PBS, and the supernatants containing laminar cells were collected. The fractions were resuspended in Percoll (GE17-0891-01, Sigma-Aldrich) after centrifugation, and the middle interface layer was harvested after density gradient centrifugation. The cell precipitate was resuspended for subsequent use. Small intestinal macrophages were cultured in 10% fetal bovine serum (FBS) and stained with APC-Cy7–conjugated anti-CD45 (I3/2.3, A15395, Thermo Fisher Scientific), BV510-conjugated anti-mouse CD11b antibody (clone M1/70, 101263, Biolegend), FITC-conjugated mouse F4/80 (clone BM8, 123108, Biolegend), and sorted.

Spleen tissues were carefully pulverized using a 70-µm cell strainer, rinsed in PBS, collected in 15-mL centrifuge tubes, and centrifuged at 400 × g for 5 min. The precipitate was resuspended in erythrocyte lysis buffer (Miltenyi Biotec, Bergisch Gladbach, Germany) and incubated for 5 min on ice. Subsequently, digestion was terminated using cold PBS and the sample was centrifuged at 400 × g for 5 min. The remaining cell pellets were resuspended within cold PBS and stored at 4 °C until further use.

After rinsing the bone marrow tissues derived from bone marrow cavities with DMEM (Gibco), the suspensions were centrifuged. Red blood cells were lysed using lysis buffer and resuspended in cold PBS for later use.

Blood samples were centrifuged at 400 × g for 15 min to obtain sera, and the remaining samples were diluted with 1 mL PBS. The diluted blood samples were then transferred gently into 15-mL centrifuge tubes containing 1 mL Ficoll Histopaque (P8900, Solarbio Life Sciences, Beijing, China). This led to the formation of two distinct layers, which were then centrifuged for 15 min at density gradients. The cell precipitate was resuspended for testing.

The cells were incubated with a protein transport inhibitor cocktail (554724, BD Biosciences) at 37 °C for 4 h and Fc block (553141, BD Pharmingen, Franklin Lakes, NJ, USA) at 4 °C for 15 min. The cells were then incubated at 4 °C for 30 min in the dark for surface staining with fluorophore-conjugated antibodies before being fixed, permeabilized, and incubated with intracellular cytokine staining antibodies against IL-10 (clone JES5-16E3) and CD206 (clone C068C2). Flow cytometry acquisition was performed using an LSRFortessa X-20 Multidimensional HD Flow Cytometer (BD Biosciences), while data analysis was performed using FlowJo software (Tree Star Inc., Ashland, Or, USA). The antibodies used are listed in Supplementary Table 2.

Small intestinal crypts were isolated from Lgr5-EGFP-IRES-CreERT2 mice, as described above. Crypts were further dissociated into single cells using TrypLE Express (Invitrogen, Waltham, MA, USA). Enhanced GFP (EGFP) stem cells were gated and sorted using a MoFlo XDP ultra-speed flow cell sorting system (Beckman Coulter, Brea, CA, USA).

## Depletion of intestinal macrophages

Two hundred microliters of clodronate- or PBS-loaded liposomes (CP-005-005, LIPOSOMA, Groningen, The Netherlands) were intravenously injected into mice thrice on alternating days prior to intestinal I/R to deplete intestinal macrophages.

## Participants

Twenty-three participants undergoing coronary artery bypass graft or elective cardiac valve replacement surgery, at the cardiac surgery department of the Nanfang Hospital of Southern Medical University (Guangzhou, China), were consecutively recruited between September 2020 and November 2021. All participants were aged between 18 and 75 years. Exclusion criteria included those with chronic digestive system diseases, previous gastrointestinal surgery, chronic kidney disease, confirmed or suspected intestinal ischemia/necrosis, and those who had used prebiotics, laxatives or antidiarrheals within one week, or antibiotics within three months before the start of the study. No significant differences between the demographic and health-related characteristics of the groups were observed. Written informed consent was obtained from all patients. Ethical committee approval of this study was obtained from the Nanfang Hospital of Southern Medical University (approval number: NFEC-202009-k2-01).

## Lausanne intestinal failure estimation (LIFE) score

The gastrointestinal failure score of patients was calculated on the seventh day after surgery in accordance with LIFE score estimation, as described previously[65]. Briefly, the evaluation criteria encompass various parameters, including levels of intra-abdominal pressure, lactate, gastric residue, enteral nutrition, motility, and bowel sounds.

## Human FABP2/I-FABP immunoassay

Human blood samples were collected preoperatively and 12 h after surgery using an anticoagulant (EDTA or heparin) and centrifuged at 400 × g for 15 min to separate the upper layer of plasma. All samples were stored at −80 °C before use. Human plasma FABP2/I-FABP levels were determined using a human FABP2/I-FABP Quantikine ELISA kit (DFBP20, R&D Systems) following the manufacturer's instructions, researchers being blinded to group allocations.

## D-lactate assay kit

Frozen human plasma samples were thawed at room temperature before detection. In accordance with the manufacturer's protocol, the colorimetric D-lactate assay kit (ab83429, Abcam) was used to measure D-lactate concentration in the plasma by researchers who were blinded to the group allocations.

## BMDM culture

To isolate BMDMs, the femur and tibiae of 7-8 week-old C57BL/6 J mice were flushed, the suspension centrifuged at 400 × g for 10 min and the supernatant discarded. Subsequently, the precipitate was resuspended in DMEM (Gibco) supplemented with 10% FBS (Gibco) as well as 20% L929 conditioned mediam and cultured in a 37 °C cell incubator. Seven days after differentiation, the cells were harvested for experiments. In the conditioned medium experiment, cells were cultured in organoid-derived conditioned medium or DMEM/F12 as a control. In the MA stimulation experiment, cells were cultured with 4 µM MA or an equal volume of double-distilled water (ddH2O) as a vehicle. Cells treated with siRNA for 24 h before stimulation with MA were used as the negative control (NC) group. Cells were randomized into different groups as follows: (i) vehicle; (ii) MA; (iii) vehicle + siNC; (iv) vehicle+ siSOCS2; (v) MA + siNC and (vi) MA + siSOCS2.

## Untargeted and targeted metabolomics

Untargeted metabolomic analysis was performed by a Vanquish UHPLC system (Thermo Fisher Scientific) and an Orbitrap Q Exactive™HF-X mass spectrometer (Thermo Fisher Scientific). To obtain extracts, samples were homogenized and repeatedly centrifuged, and the final fecal extracts were aliquoted for metabolite profiling analyses. Raw data was generated using ultra-high performance liquid chromatography-mass spectrometry and processed by

Compound Discoverer 3.1 (CD3.1, Thermo Fisher Scientific) for integration, normalization, and peak intensity alignment. Next, mzCloud, mzVault, and MassList databases were used to match the processed dataset. Principal component analysis and OPLS-DA were performed using normalized data, and VIP > 1 was considered as the threshold.

Targeted quantitative metabolomics was conducted using TSQ Quantiva™ (Thermo Fisher Scientific). The MA standard (L46691, Acmec, Biochemical, Shanghai, China), αKG standard (61234, Sigma-Aldrich) were dissolved in 4:1 methanol/water (v/v) to prepare the standard curve. The samples were homogenized with 80% methanol, sonicated for 3 min, and centrifuged at 14,000 × g at 4 °C for 10 min. The supernatant obtained was filtered for subsequent experiments. A Prelude SPLC™ sample preparation and liquid phase system (Thermo Fisher Scientific) was used to carry out Chromatographic separation, TSQ Quantiva (Thermo Fisher Scientific) was used to perform quantitative detection and the TraceFinder™ software version 3.3 SP1 (Thermo Fisher Scientific) was used to complete data collection.

### RNA-seq

RNA was extracted as mentioned above. RNA sequencing was performed by Shanghai Biotechnology Corporation. Briefly, all RNA samples underwent quality control analysis using Fast QC (version 0.10.1). Sequencing libraries were prepared using a TruSeq Stranded Total RNA Library Prep Kit (15032612, Illumina, San Diego, CA, USA). CBot (HiSeq2500) was used to generate clusters and the HiSeq2500 sequencing platform (Illumina) was used to generate pair-end 150 bp reads. Valid clean data were obtained using Trimadap to filter out unqualified sequences, while HISAT2 was used for reference genome alignment. Based on HISAT2 alignment results, transcripts were reconstructed using StringTie (John Hopkins University, Baltimore, MD, USA), and expression levels of all genes in each sample were calculated. The computation of differential gene expression was completed by the negative binomial model implemented in the Bioconductor package, edgeR. A fold change (FC) ≥ 2 or FC ≤ -2 with a q-value < 0.05 was considered significant.

### siRNA transfection

Mouse siRNA targeting SOCS2 and negative control siRNA were designed and constructed by Ribo Targets (Guangzhou, China). Transfection was performed via Lipofectamine 3000 (L3000015, Invitrogen), following the manufacturer's instructions. The efficiency of SOCS2 knockdown was assessed using qRT-PCR and western blotting.

**Macrophage isolation.** Donor mice were gavaged with MA for 5 consecutive days and subjected to intestinal I/R injury for 36 h. Then the mice were sacrificed and the spleen was harvested. Spleen immune cell was isolated as described above and macrophages were further isolated by using magnetic bead separation methods. In short, determine the cell number in the single cell suspension and centrifuge cell suspension. Incubate the cell pellet with anti-F4/80 microbeads (130-110-443, Miltenyi Biotec, Germany) according to the manufacturer's instructions.

**Adoptive transfer.** Recipient mice underwent intestinal I/R injury were injected intravenously with 2-3 × 10^6 macrophages immediately after the clamp was released.

### Statistical analysis

Statistical analyses were performed using GraphPad Prism (version 8.3.0) software. Two-tailed log-rank test was used for the Kaplan-Meier survival curve. The Shapiro-wilk test is used to determine whether the data is normally distributed. For normally distributed data, two-tailed student's *t* test was used to compare the means of two independent groups. One-way ANOVA and Two-way ANOVA, followed by Tukey's multiple comparisons test, were used to compare the means of more than two groups. For non-normally distributed data, the nonparametric method such as Mann–Whitney test was used. All data are expressed as mean ± standard deviation. P < 0.05 was considered statistically significant. For further statistical details, refer to individual figure legends.

### Reporting summary

Further information on research design is available in the Nature Portfolio Reporting Summary linked to this article.

## Data availability

All data are available in the main text or the supplementary materials. The differentially expressed raw metabolomic data have been provided in Supplementary Data. The raw RNA–seq data for this study have been deposited in the Sequence Read Archive (SRA) under accession code SRP434172. Source data are provided with this paper.

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

## Acknowledgements

The figure schematics were made using Smart Medical Art (SMART): https://smart.servier.com/. The authors express gratitude to Professor Peng Chen for valuable guidance and Professor Zai-Long Chi for providing SOCS2$^{-/-}$ breeding pairs. Thanks to the funding support from National Natural Science Foundation, Beijing, China (82172141 to Kexuan Liu), Key Program of National Natural Science Foundation, Beijing, China (82330067 to Kexuan Liu).

## Author contributions

K.-X.L., F.-L.Z. and Z.H. contributed to the design and administration of the project. F.-L.Z. and Z.H. contributed to the experiment's performance. Y.-F.W. collected the experiments data. Z.-B.L. and Q.-S.S. collected fecal and blood samples from patients. F.-L.Z. and Y.-F.W. checked the accuracy of the data and performed the statistical analysis. F.-L.Z. and W.-J.Z. assembled tables and figures. B.-W.Z. edited and reviewed the manuscript. All authors revised the manuscript. The authors read and approved the final manuscript.

## Competing interests

The authors declare no competing interests.
