## [Peer Review File · Nature Communications]

REVIEWER COMMENTS

Reviewer #1 (Remarks to the Author):

In the manuscript “Organoids transplantation attenuates intestinal ischemia/reperfusion injury through L-Malic acid-mediated M2 macrophage polarization” the authors present an elegant approach to improve recovery from intestinal IR injury using intestinal organoids. They find that intestinal organoid-based therapy improves survival of mice exposed to intestinal IR injury, and they investigated the mechanisms that are responsible for the success of this treatment. The data are in general of good quality, technically sound and are produced using appropriate methodology. The conclusions are supported by the data, results are clearly described, the evidence is strong, and authors made an attempt to provide some clinical relevance. Data are of significance and can lead to improved therapy for conditions in which IR injury plays a role.

The data are original as no other study showed the use of organoids to improve intestinal IR injury, however, organoid-based therapy is already used successfully in other conditions of digestive diseases. In addition, a protocol is published (Nat Protoc 2022;17(3):649-671) showing how to transplant intestinal organoids into a mouse model of colitis.

Some comments to improve the manuscript are listed here:

Title should include “in mice”

Materials and methods:

Authors should use the ARRIVE guidelines to describe the animal research performed for obtaining the data reported in this manuscript. Also, data on animal welfare monitoring, pain management etc. should be included, as well as how survival was monitored. Authors should report on the method of randomization and blinding etc.

Some detail on the orthotopic transplantation is lacking. Organoids were administered intraduodenally. How many injections were given over the entire ischemic segment? How long was the ischemic segment? How were control mice treated? These details should be included.

Authors should provide some more detail on how crypt depth was measured as well as the degree of intestinal injury. Line 566 mentions Chiu’s method with modifications.

Data on ZO-1 and Occludin staining seem to be primarily determined by the mucosal surface that is present and not so much on how effective tight junctions are. Figure 3E and 3F support this. If authors want to have a clear readout with regard to intestinal barrier function, they could use e.g. FITC-dextran method.

Statistics:

Was a normality check done before performing students t-test?

Data should be presented as mean/SD and not SEM.

Results:

In the legend of all figures, the number of animals used per experimental group should be mentioned.

E.g. Figure 1A. How many mice per group were used for the survival analysis?

Figure 2 B, C, D does not fit within the subject of Figure 2, which is "The myeloid landscape." These figures fit better with content of Figure 1.

Figure 6H. IL10 data are confusing. It is expected that IL10 expression is lower in both siSOCS2 treatment groups. The vehicle + siSOCS2 group has the highest IL10 expression in this figure and not the MA + siNC group.

Authors should improve readability of some labels (of X/Y axes), names etc. in the figures. E.g. labels of supplementary Figure 2, metabolite names in Fig 4H, text Fig 6B, the symbols (squares and bullets) of Fig 6I.

Authors should provide the complete list of differentially expressed metabolites of organoid conditioned medium and cecal content, as well as all data obtained gene expression data (RNAseq data) in valid repositories such as the Gene Expression Omnibus (GEO).

Fig1B and Fig7 mentions "donar" mice instead of "donor" mice.

Discussion:

Data of Figure 1 are a bit puzzling; they show only a significant difference between treatments in most outcome parameters on injury, inflammatory mediators, proliferation etc. at 36h after ischemic injury

while death is primarily taking place early in the postoperative course, between 5 and 30 hours. How can authors explain this apparent discrepancy?

Clinical relevance:

Authors should refer to the transplantation protocol that has been recently published and explain in what way the current protocol deviated from the published protocol in *Nat Protoc* 2022;17(3):649-671 and what this means in terms of clinical applicability.

The authors should discuss the potential clinical implications a bit better. How could patients benefit from this therapy. Intestinal IR is an acute condition that needs immediate treatment. How could organoids be of help here?

Data in figure 5 suggest MA treatment is as efficient as organoid transplantation, while this is a much simpler approach. Do authors believe the organoid-based therapy has a real clinical potential for intestinal IR, or might MA administration be sufficient? And how could this be administered?

Reviewer #2 (Remarks to the Author):

Development of intestinal organoid cultures has opened the possibility for transplantation to restore the epithelial barrier in cases of mucosal injury. However, the cellular and molecular mechanisms underlying the possible beneficial effects of intestinal organoid transplantation are yet to be elucidated. In the current study, Fang-Ling Zhang Zhen Hu and colleagues showed that organoids transplanted into mice with intestinal ischemia reperfusion (I/R) injury resulted in improved survival, regulated the immune microenvironment, and promoted stem cell self-renewal. The authors proposed that this was due to the enhanced tissue-protective activity of macrophages polarized to an anti-inflammatory phenotype, which was induced by L-malic acid secreted during engraftment.

While this study provides a possible new therapeutic strategy for treating intestinal I/R injury, a more mechanistic explanation should be provided to define the role of L-malic acid and macrophages.

The major point of discussion here is the role of macrophages and their polarization state in an inflamed gut. The authors proposed that macrophages acquire an anti-inflammatory phenotype under the influence of L-malic acid.

- Here the authors are not taking into account that probably reconstitution of the epithelial barrier via the intestinal organoid transplantation would already in turn have a beneficial effect on macrophages?

-It is also not clear whether the beneficial effect of L-malic acid is mediated via an effect on tissue-resident macrophages (increased WNT signalling) or by promoting differentiation of newly recruited monocytes. This possibility should be experimentally dissected by following changes in the molecular and cellular architecture of mucosal myeloid cells during the course of I/R injury upon transplantation.

-Depletion of macrophages using liposomes results in large alterations in myeloid cellular compartments in different organs. In addition, macrophage and monocyte depletion achieved with clodronate liposomes leaves the intestinal mucosa unarmed against bacteria crossing the damaged epithelial barrier, possibly delaying tissue recovery.

-The source of l-malic acid and the mechanism leading to its increased secretion from intestinal organoids have not been described and discussed.

-The cellular and molecular targets of l-malic acid have not been identified in vivo. The direct effect of l-malic acid should be fully demonstrated in vivo using specific SOCS2 macrophage-deficient mice. In addition, to fully demonstrate the potential clinical applicability of this finding, the effect of L-malic acid could be demonstrated independently of organoid transplantation via pharmacological administration in a preclinical I/R injury model

Reviewer #3 (Remarks to the Author):

Overall impression

The manuscript is well written, and clearly illustrates the biological effect of successful organoids transplantation via L-Malic secretion from organoids and its downstream signal cascade. The importance of IL-10 to suppress STAT3 phosphorylation is also well documented with clear results. Not only transplantation system but also culture system using conditioned medium of intestinal organoids are quite unique systems, which would benefit the related field. Particularly this reviewer agreed that organoids transplantation can even modify the systemic immunity based on the content of Figure 2. This part is a quite novel aspect of organoids transplantation, which may be beneficial for a broad range of readers in various research fields. The process to target L-Malic acid is also well-documented and clear.

As one of the experts in the field, this reviewer may have some comments as described below. Please consider take these issues into the manuscripts.

Comments

1. In page 3 in the introduction, the manuscript explains about organoids transplantation citing some of the important references. This reviewer thinks that the following paper might be a candidate of reference here. Satoshi Watanabe et al, Nature Protocols, 2022 (<https://doi.org/10.1038/s41596-021-00658-3>). Please consider whether this citation is correct one in references.
2. This reviewer thinks that immunofluorescence for Ki67 in Figure 1I and Supplementary Figure 4A is not so clear. Hence the valid quantification supports the conclusion, this reviewer prefers the higher magnification. Please consider which is better.
3. In page 7 in the sentence 'Notably, our results are consistent with previous study of higher engraftment rate of small intestinal organoids grown from ...', this reviewer thinks that this is 'colonic' instead of 'small intestinal'.
4. In the same paragraph of page 7, there is a sentence 'we found a scattered distribution of donor organoids in the recipient's small intestinal', but this reviewer thinks this is 'small intestine' rather than 'small intestinal'.
5. This reviewer could not find the call for Figure 7.
6. In the discussion part, the authors discuss the reason of higher engraftment rate in organoids compared to sorted Lgr5 cells or crypts. This reviewer thinks that amplification of stem cells fraction during in vitro culture might be one of reasons, and this might be better to include this with appropriate references.

Response letter

Reviewer #1 (Remarks to the Author):

In the manuscript “Organoids transplantation attenuates intestinal ischemia/reperfusion injury through L-Malic acid-mediated M2 macrophage polarization” the authors present an elegant approach to improve recovery from intestinal IR injury using intestinal organoids. They find that intestinal organoid-based therapy improves survival of mice exposed to intestinal IR injury, and they investigated the mechanisms that are responsible for the success of this treatment. The data are in general of good quality, technically sound and are produced using appropriate methodology. The conclusions are supported by the data, results are clearly described, the evidence is strong, and authors made an attempt to provide some clinical relevance. Data are of significance and can lead to improved therapy for conditions in which IR injury plays a role.

The data are original as no other study showed the use of organoids to improve intestinal IR injury, however, organoid-based therapy is already used successfully in other

conditions of digestive diseases. In addition, a protocol is published (Nat Protoc 2022;17(3):649–671) showing how to transplant intestinal organoids into a mouse model of colitis.

Response to general comment: We sincerely thank the reviewer for the helpful suggestions to improve data transparency and clarity of the manuscript. Below please find a point-by-point response to the comments. Specifically,

- 1) We have provided more detailed information on the Materials and Methods section for better clarity and reproducibility.
- 2) We have provided the raw metabolomic data along with the accession code for the raw RNA-seq data to ensure data transparency.
- 3) We have further clarified the potential clinical implications of organoid transplantation for clinical transformation.

Some comments to improve the manuscript are listed here:

Title should include “in mice”

Response 1: We thank the reviewer for this suggestion, we updated the manuscript title to “Organoids transplantation attenuates intestinal ischemia/reperfusion injury in mice through L-Malic acid-mediated M2 macrophage polarization”.

Materials and methods:

Authors should use the ARRIVE guidelines to describe the animal research performed for obtaining the data reported

in this manuscript. Also, data on animal welfare monitoring, pain management etc. should be included, as well as how survival was monitored. Authors should report on the method of randomization and blinding etc.

Response 2: We thank the reviewer for the detailed comments regarding the inadequate description in the Materials and methods section. We rearranged our manuscript and added clarification within the manuscript about the requested details so that the manuscript may comply with the ARRIVE guidelines. Clarifications included how animal welfare was monitored, method of randomization, blinded nature of the study, and additional details regarding pain management were added to the manuscript as well in materials and methods section in the revised manuscript:

Materials and Methods

Materials and Methods (Line 468): “Mice with the same genotype were assigned randomly to each group by an independent experimenter, but no specific randomization method was employed. Additionally, no statistical methods were utilized to pre-estimate the sample size. Experimenters responsible for histological analyses, IF and relevant laboratory tests were blinded to the grouping of the mice”.

Animals (Line 485): “All animal handling were according to the policy of the China Animal Welfare guidelines and under the supervision of the ethics committee of Southern Medical University”.

Establishment of the mouse intestinal I/R injury model (Line 505): “After the abdominal incision was closed with suture, 1 mg/ml butorphanol (Jiangsu Hengrui Medicine Co., Ltd., Jiangsu, China) was injected subcutaneously for pain management”.

Establishment of the mouse intestinal I/R injury model (Line 507): “The survival of the mice was monitored every 6 hours after releasing the artery clamp and continued to be monitored for a study period of 7 days”.

Some detail on the orthotopic transplantation is lacking.

Organoids were administered intraduodenally. How many injections were given over the entire ischemic segment? How long was the ischemic segment? How were control mice treated? These details should be included.

Response 3: We sincerely thank the reviewer for this valuable suggestion for improvement. The gastrointestinal tract is mainly composed of the celiac artery, superior mesenteric artery, and inferior mesenteric artery for blood supply (Radiol Clin North Am. 2010;48(2):331-viii). The superior mesenteric artery mainly supplies the jejunum and ileum as well as the splenic curvature of the colon (Compr Physiol. 2015;5(3):1541-1583). Consistent with previous reports, our research to be published also found that the clamping of the superior mesenteric artery in mouse model causes almost full-length injury of the small intestine (**Author Response Figure 1**).

Author Response Figure 1. Representative H&E images and quantification of the histopathology changes of different intestinal tissue segments (n=4-5).

Therefore, in this study, we have used only one injection intraduodenally in our experiments. We followed the suggestion of the reviewer and added these contexts in the Materials and Methods section as bellows:

Materials and Methods

Orthotopic transplantation (Line 546): “200 μ L organoid solution was injected once into the duodenal lumen from the proximal to distal portion of the small intestine using a 1-mL syringe with a 23-gauge needle and the control DMEM/F12 and Matrigel without organoids were injected in the same manner”.

Authors should provide some more detail on how crypt depth was measured as well as the degree of intestinal injury.

Line 566 mentions Chiu’ s method with modifications.

Response 4: We thank the reviewer for the valuable suggestion. We have added the details for the measurement of crypt depth and intestinal injury scores in the Materials and Methods section which reads:

Materials and Methods

Histology and immunohistochemistry (Line 584): “In brief, the modification pathological scoring criteria are as follows: 0, normal intestinal mucosa and intestinal villus. 1, formation of Gruenhagen cavity begins at the tip of the intestinal villus. 2, formation of Gruenhagen cavity and slight damage of glands. 3, formation and enlargement of subepithelial gaps, congested and engorged capillaries. 4, moderate detachment of epithelium from lamina propria and gland damage. 5, partial loss of villus at the tip. 6, obvious loss of villus and dilated capillaries. 7, loss of lamina propria and significant gland damage. 8, beginning of digestion and decomposition of the lamina propria. 9, hemorrhage and formation of ulcers. The depth of the crypt was measured from the bottom to the opening”.

Data on ZO-1 and Occludin staining seem to be primarily determined by the mucosal surface that is present and not so much on how effective tight junctions are. Figure 3E and 3F support this. If authors want to have a clear readout with regard to intestinal barrier function, they could use

e. g. FITC-dextran method.

Response 5: We thank the reviewer for pointing this out and agree that FITC-dextran method could enhanced the intestinal barrier permeability more effectively. Tight junctions are the primary determinants of barrier function in intact epithelia and correlated with intestinal permeability (Nat Rev Gastroenterol Hepatol. 2017;14(1):9-21.). The tight junctions, multiple protein complexes, locate at the apical ends of the lateral membranes of intestinal, including occludin, claudins, and zonula occludens (ZO-1) (Cell Mol Life Sci. 2013;70(4):631-659.), are crucial for the maintenance of epithelial barrier integrity. Consistent with previous studies (Immunity. 2015;43(4):727-738., Gastroenterology. 2006;131(4):1153-1163.), our results in **Manuscript Figure 5C and 5D** shown that the reduced tight junction protein expression in the vehicle group represents the injured intestinal barrier integrity and correlated with higher intestinal permeability.

Following the reviewer's suggestion, we performed additional experiments on the permeability of intestinal barrier function by measurement paracellular passage of FITC-dextran, we found that the mice in the transplantation group exhibited lower intestinal barrier permeability compared to the control group (**Author Response Figure 2A**). Additionally, the administration of MA significantly reduced serum FITC-dextran levels compared to the vehicle group (**Author Response Figure 2B**). These results suggested that organoids transplantation and MA administration help maintain the intestinal homeostasis.

Author Response Figure 2. (A-B) Intestinal permeability assessed by serum FITC-dextran level at indicated groups (n=3).

Statistics:

Was a normality check done before performing students t-test?

Data should be presented as mean/SD and not SEM.

Response 6: We thank the reviewer for this suggestion. We checked the normality of data by Shapiro-Wilk normality test before performing students t-test analysis. Following the reviewer's suggestion, we reorganized all the statistical graph to present the data as mean \pm SD (**Manuscript Figure 1-7, Supplementary Figure 1-4**). We added these descriptions to the revised manuscript which reads:

Materials and Methods

Materials and Methods (Line 785): "The Shapiro-wilk test was used to determine whether the data is normally distributed. For normally distributed data, two-tailed student's t-test was used to compare the means of two independent groups. One-way ANOVA and Two-way ANOVA, followed by Tukey's multiple comparisons test, were used to compare the means of more than two groups. For non-normally distributed data, the nonparametric method such as Mann-Whitney test was used. All data are expressed as mean \pm standard deviation".

Results:

In the legend of all figures, the number of animals used per experimental group should be mentioned.

E.g. Figure 1A. How many mice per group were used for the survival analysis?

Response 7: We apologize for the lack of clarity in the figure legends. In the revised manuscript, we have included the sample size of each group for animal and cellular experiments.

Figure legends

Figure 1B (Line 994): “Kaplan-Meier survival curve of organoid-transplanted and control mice after induction of intestinal I/R (n = 17)”.

Figure 2 B, C, D does not fit within the subject of Figure 2, which is “The myeloid landscape.” These figures fit better with content of Figure 1.

Response 8: At the reviewer’s suggestion, we have adjusted the Figure 2D to revised **Manuscript Figure 1C** and removed the Figure 2B, C due to the space limit in Figure 1. We have re-adjusted the order of some sentences to make the description of this study more logical in the revised manuscript which reads:

Results (Line 94): “to investigate the effects of organoids transplantation during different stages of intestinal inflammation, the mice were subjected to 50 min of ischemia and then transplanted with organoids or control solution and sacrificed at indicated time points (Figure 1A). Next, we assessed the effect of organoids engraftment on mouse survival seven days after intestinal I/R injury. Transplantation of the organoids significantly reduced mortality compared with that in the control group (Figure 1B). We further confirmed the successful engraftment of organoids through immunofluorescence which revealed the organoids grown from Lgr5-eGFP-IRES-CreERT2 were attached and incorporated into the injured epithelial mucosa (Figure 1C). Notably, our results are consistent with previous study¹¹ of higher engraftment rate of cultured organoids, rather than freshly isolated crypt or sorted leucine-rich repeat-containing G-protein coupled receptor 5 (LGR5)⁺ stem cells 36 h after intestinal I/R injury (Supplementary Figure 1A-D)”.

Figure 6H. IL10 data are confusing. It is expected that IL10

expression is lower in both siSOCS2 treatment groups. The vehicle + siSOCS2 group has the highest IL10 expression in this figure and not the MA + siNC group.

Response 9: We thank the reviewer for this point and apologize for this unintended oversight. We checked the raw data and found the color of the figure caption was incorrectly associated with the grouping name during the process of making figure. To confirm our conclusion, we re-performed the flow cytometry analysis to quantify IL-10 expression in the indicated groups. The new data clearly demonstrated that following siSOCS2 pretreatment, there is no difference in IL-10 expression between the vehicle + siSOCS2 group and the MA + siSOCS2 group (**Author Response Figure 3, Manuscript Figure 6F**), whereas the administration of MA significantly increased IL-10 expression. We have updated these data in the revised manuscript.

Author Response Figure 3 (Manuscript Figure 6F). Representative histograms of IL-10 expression and quantification of IL-10⁺ MFI (n = 3).

Authors should improve readability of some labels (of X/Y axes), names etc. in the figures. E.g. labels of supplementary Figure 2, metabolite names in Fig 4H, text Fig 6B, the symbols (squares and bullets) of Fig 6I.

Response 10: We appreciate this suggestion and agree that the original text fonts were too small to read. In the revised manuscript, we have simplified the main figures and enlarged their fonts to ensure readability. Moreover, we have standardized the thicknesses of Axis labeling to 1pt and font size to Arial 6pt for the entire documents. Specifically, we have enlarged the label thicknesses of **Supplementary Figure 1** to 1pt. Furthermore, we have enlarged the font size to Arial 6pt in the revised **Manuscript Figure 4F and 6A**. The symbols style of revised **Manuscript Figure 6F** have been adjusted for different color which may help improving the readability and make the data look clearer.

Authors should provide the complete list of differentially expressed metabolites of organoid conditioned medium and cecal content, as well as all data obtained gene expression data (RNAseq data) in valid repositories such as the Gene Expression Omnibus (GEO).

Response 11: We thank the reviewer for this suggestion. The complete list of differentially expressed metabolites of organoid conditioned medium and cecal content are provided in attached file. The raw RNA-seq data for this study have been deposited in the Sequence Read Archive (SRA) under accession code SRP434172. We added this information to the revised manuscript which reads:

Data availability (Line 817): “The raw RNA-seq data for this study have been deposited in the Sequence Read Archive (SRA) under accession code SRP434172”.

Fig1B and Fig7 mentions “donar” mice instead of “donor” mice.

Response 12: We thank the reviewer for catching this. We apologize for this overlook and have corrected ‘donar’ as ‘donor’ in the revised **manuscript Figure 1A and Figure 8**.

Discussion:

Data of Figure 1 are a bit puzzling; they show only a significant difference between treatments in most outcome parameters on injury, inflammatory mediators, proliferation etc. at 36h after ischemic injury while death is primary taking place early in the postoperative course, between 5 and 30 hours. How can authors explain this apparent discrepancy?

Response 13: We thank the reviewer for this insightful comment. We discovered that the expression of serum IL-10 in the transplanted group increased significantly over time when compared to the control group (**Manuscript Figure 1F**). This cytokine is known for its anti-inflammatory properties and can promote tissue repair and regeneration (Immunity. 2019;50(4):871-891.). Previous studies have also demonstrated that IL-10 administration significantly reduced the early mortality rate of mice in the transverse aortic constriction model, while cardiac function gradually improves in the middle and late stages (Circulation. 2012;126(4):418-429.), which indicated that there may be discrepancy in function recovery and mortality rate. Moreover, our data shows that neutrophils and monocytes significantly increased in the control group as compared to the transplanted group at 6 h and 12 h after intestinal I/R injury (**Manuscript Figure 2A**). This could be attributed to the fact that certain outcome parameters take faster to manifest and achieve significant levels. We hope the above analyses and clarification help at least partially explain the discrepancy between some outcome parameters and the early mortality rate.

Clinical relevance:

Authors should refer to the transplantation protocol that has been recently published and explain in what way the

current protocol deviated from the published protocol in Nat Protoc 2022;17(3):649–671 and what this means in terms of clinical applicability.

Response 14: We thank the reviewer for pointing out that some necessary references in the manuscript are missing. Series of studies by Shiro Yui et al. has provided an apparent breakthrough in this area, by their establishment of a novel method for the orthotopic transplantation of organoids (Nat Med. 2012;18(4):618-623.). The major distinction between the current protocol and the published protocol in Nat Protoc 2022;17(3):649-671 is that the target region of transplantation. In the model of intestinal I/R injury, the injury occurs in the small intestine. Consequently, we administered the transplantation of organoids intraduodenally in our experiments. Moreover, taking into account the peristaltic function of the small intestine, we adjusted the optimal ratio of matrigel to solvent for improved adhesion. Currently, the effective approaches for intestinal I/R injury clinical application are still lacking. Moreover, organoids transplantation has shown certain therapeutic effects in various intestinal diseases (Genes Dev. 2014;28(16):1752-1757., Nature. 2021;592(7852):99-104.). Therefore, we referred to the transplantation methods mentioned in the literature and modifying the protocol to better fit the mouse intestinal I/R model and ensure successful organoid transplantation.

As suggested, we have now discussed the differences of the current protocol and the published protocol in Nat Protoc 2022;17(3):649-671 in the revised manuscript with the relevant citation added.

Discussion (Line 375): “Additionally, we have modified the delivery route of the organoids and optimized the ratio of Matrigel to solvent to better fit the intestinal I/R model and ensure the successful transplantation of organoids. This method has the potential to benefit patients through minimally invasive techniques in the future application. Moreover, the standardization protocol for intestinal organoids transplantation and future extensive research will contribute to its application in the treatment of human diseases⁴⁴”.

The authors should discuss the potential clinical implications a bit better. How could patients benefit from

this therapy. Intestinal IR is an acute condition that needs immediate treatment. How could organoids be of help here?

Response 15: We thank the reviewer so much for raising these important points. We initiated organoids transplantation immediately after the clamp released, which is more suitable for clinical transformation in a postconditioning way rather than preconditioning (**Manuscript Figure 1A**). The high mortality rate among patients with intestinal I/R injury in clinical practice is due to extensive necrosis of intestinal tissue and multiple organ dysfunction (Can Assoc Radiol J. 2023;74(1):160-171.). Hence, there is an urgent need to develop and test new treatment strategies to improve this condition. Stem cell transplantation is a potential method, but it is limited by various factors (Circ Res. 2013;113(3):288-300., Nat Rev Neurosci. 2006;7(8):628-643.). Nevertheless, unlike stem cell transplantation, our results show that organoid transplantation has a higher engraftment efficiency, making it a promising biomimetic transplantation material. Furthermore, intestinal I/R injury leads to tissue necrosis and an overactivated immune system, resulting in severe inflammatory responses and organ dysfunction. Our results suggest that transplanted organoids could modulate the immune microenvironment and alleviated the inflammation. In future application, we are hopeful that modifying anti-inflammatory organoids patches could better regulate patients' immune systems, reduce inflammatory responses, and promote tissue repair and recovery. Moreover, the patients could benefit from organoids transplantation relies on minimally invasive techniques, which reduces the need for complex surgeries, resulting in shorter recovery times, fewer complications, and decreased healthcare costs. Furthermore, organoids can be stably acquired and expanded from a single biopsy-sized epithelium, which can provide an innovative and ethical solution. As suggested, we added this information to the revised manuscript which reads:

Discussion (Line 374): "Here we have developed a postconditioning transplantation method for intestinal I/R injury that is more applicable for clinical transformation. Additionally, we have modified the delivery route of the organoids and optimized the ratio of matrigel to solvent to better fit the intestinal I/R model and ensure the successful transplantation of organoids. This method has the potential to benefit patients through minimally invasive techniques in the future application. Moreover, the standardization protocol for intestinal organoids transplantation and future extensive research will contribute to its application in the treatment of human diseases⁴⁶".

Data in figure 5 suggest MA treatment is as efficient as organoid transplantation, while this is a much simpler approach. Do authors believe the organoid-based therapy has a real clinical potential for intestinal IR, or might MA administration be sufficient? And how could this be administered?

Response 16: We thank the reviewer for addressing this important point. Organoids transplantation induced the repair process by engraftment and cell replacement which cannot be simply replaced by MA administration. Additionally, multiple metabolites, functional enzymes, growth factors, and hormones are secreted by mature organoids, indicate the intricate paracrine signal capabilities and potential therapeutic benefits of organoids. Most importantly, the successful transplanted organoids can continuously provide various cytokines, which cannot be simply replaced by MA administration. Thus, we believe that the therapeutic potential of organoids therapy is much more than administration of MA. We have discussed this limitation in the revised manuscript which reads:

Discussion (Line 428): "However, multiple functional enzymes, growth factors, and hormones are secreted by mature organoids. Thus, further analyses of the protein profile of organoid-derived conditioned media may provide more therapeutic options for future applications²³⁻²⁵".

Reviewer #2 (Remarks to the Author):

Development of intestinal organoid cultures has opened the possibility for transplantation to restore the epithelial barrier in cases of mucosal injury. However, the cellular and molecular mechanisms underlying the possible beneficial effects of intestinal organoid transplantation are yet to be elucidated. In the current study, Fang-Ling Zhang Zhen Hu and colleagues showed that organoids transplanted into mice with intestinal ischemia reperfusion (I/R) injury resulted in improved survival, regulated the immune microenvironment, and promoted stem cell self-renewal. The authors proposed that this was due to the enhanced tissue-protective activity of macrophages polarized to an anti-inflammatory phenotype, which was induced by L-malic acid secreted during engraftment.

While this study provides a possible new therapeutic strategy for treating intestinal I/R injury, a more mechanistic explanation should be provided to define the role of L-malic acid and macrophages.

The major point of discussion here is the role of macrophages

and their polarization state in an inflamed gut. The authors proposed that macrophages acquire an anti-inflammatory phenotype under the influence of l-malic acid.

Response to general comment: We thank the reviewer for the thoughtful comments along with the many helpful suggestions to improve the manuscript. We have attempted to address the reviewer's concerns by performing additional experiments, analyses and revisions. Specifically,

1) We have sorted intestinal macrophages in mice and attempted to explore the involved signaling pathway under MA administrated.

2) We have constructed the SOCS2 knocked-out mice to determine whether MA controls macrophage polarization and intestinal recovery from injury in a SOCS2-dependent manner in vivo.

3) We have added the discussion for the source of MA and the mechanisms leading to the high metabolic level in organoids as suggested.

- Here the authors are not taking into account that probably reconstitution of the epithelial barrier via the intestinal organoid transplantation would already in turn have a beneficial effect on macrophages?

Response 1: We thank the reviewer for the valuable suggestions. The reviewer is correct that intestinal organoids transplantation was able to engraft in areas of ulceration which may provide a physical and biochemical barrier (Genes Dev. 2014;28(16):1752-1757.). As the reviewer indicated that the integrity of intestinal barrier structure is closely related to the function of macrophages. Specifically, the immunoregulatory functions of IECs, their contribution to the priming of adaptive immune cell responses, regulation of innate effector responses and homeostasis of adaptive immune cell function in the intestinal environment to some extent depends on numerous immunoregulatory signals they produced (Nat Rev Immunol. 2014;14(3):141-153., Trends Immunol. 2011;32(6):256-264.), which was confirmed in our research through conditional medium experiments and Liquid Chromatograph Mass Spectrometry analysis.

Additionally, although the tight junction protein level of Occludin and ZO-1 were higher in the transplanted group compared to the control group after 36 hours of intestinal I/R injury, they were both significantly lower than the levels observed in the sham group (**Author Response Figure 4A, B, Manuscript Supplementary Figure 2B, C**), which indicated incomplete mucosal recovery of intestinal in 36 hours after intestinal I/R injury. Considering the various modes of crosstalk exist between the epithelial barrier and the immune microenvironment, we have toned down the statements and added more discussion in the revised manuscript which reads:

Author Response Figure 4 (Manuscript Supplementary Figure 2B, C). Quantification of the area of Occludin and ZO-1 immunohistochemistry staining at indicated groups.

Discussion (Line 403): “The integrity of intestinal barrier structure is closely related to the function of macrophages through various ways ⁴⁹. Specifically, the immunoregulatory functions of epithelium, their contribution to the regulation of innate and adaptive immune responses in the intestinal environment to some extent depends on numerous immunoregulatory signals they produced ^{49,50}. Therefore, we hypothesized that organoids may function via a secretory mechanism influenced the immune microenvironment”.

-It is also not clear whether the beneficial effect of L-malic acid is mediated via an effect on tissue-resident macrophages (increased WNT signalling) or by promoting differentiation of newly recruited monocytes. This possibility should be experimentally dissected by following

changes in the molecular and cellular architecture of mucosal myeloid cells during the course of I/R injury upon transplantation.

Response 2: We thank the reviewer for raising this thoughtful comment. We performed additional experiments to address these issues. As suggested, we investigated the WNT signaling pathway in the sorted lamina propria macrophages from L-malic acid (MA) administrated mice and vehicle mice. We found that there was no significant difference on the expression of Wnt3 and its co-receptor Lrp5 between macrophages in the MA administrated group and the vehicle group. Moreover, there was no significant difference in the expression of Wnt canonical modulator Axin2 and other key target genes (SOX9, TCF-1) between macrophages in both groups (**Author Response Figure 5**) which is consistent with the findings of RNA sequencing (RNA-seq) analysis conducted on MA administration in vitro (**Manuscript Figure 6B**).

Author Response Figure 5. Gene expression of canonical Wnt signaling target genes in macrophages sorted from mice intestinal lamina propria following treatment with MA or vehicle.

To further assess the effect of MA on promoting recruited monocytes differentiation, we identified the three subsets of monocytes during their recruitment and differentiation into macrophages in the intestinal lamina propria (LP). We observed that MA can reduced the proportion of inflammatory Ly6C⁺MHC II⁻ monocytes and increased the proportion of monocyte-derived macrophages, including Ly6C⁺MHC II⁺ (**Author Response Figure 6, Manuscript Figure 7E**). These results providing further evidence that MA could promoted the differentiation of monocytes. These results are presented in **Manuscript Figure 7E** in the revised manuscript.

Author Response Figure 6 (Manuscript Figure 7E). Expression of Ly6C and MHC II by live CD45⁺CD11b⁺ small intestinal LP cells in mice 36 h after intestinal I/R.

-Depletion of macrophages using liposomes results in large alterations in myeloid cellular compartments in different organs. In addition, macrophage and monocyte depletion achieved with clodronate liposomes leaves the intestinal mucosa unarmed against bacteria crossing the damaged epithelial barrier, possibly delaying tissue recovery.

Response 3: We thank the reviewer for raising an important point regarding the possibility of delayed tissue recovery by depleting macrophages. In order to address this concern, we conducted an in vivo macrophages adoptive transfer assay to investigate the function of macrophages in intestinal I/R injury.

To further dissect the function of MA-treated intestinal macrophages, we conducted an adoptive transfer assay. Macrophages were sorted from MA-administrated WT or SOCS2^{-/-} mice undergoing intestinal I/R injury for 36 h. Recipient WT and SOCS2^{-/-} mice were administrated with adoptive transfer after the intestinal ischemia (**Author Response Figure 7A, Manuscript Figure 7H**). We observed less intestinal pathological damage in both WT mice and

SOCS2^{-/-} mice upon the adoptive transfer of the macrophages from MA-administrated WT mice (**Author Response Figure 7B, Manuscript Figure 7I**). In line with this result, adoptive transfer of the macrophages from MA-administrated WT mice restored intestinal barrier integrity (**Author Response Figure 7C, Manuscript Figure 7J**) and IL-10 secretion (**Author Response Figure 7D, Manuscript Figure 7K**), decreased the serum IL-6 and IL-1 β level, whereas adoptive transfer of the macrophages from MA-administrated SOCS2^{-/-} mice did not (**Author Response Figure 7D, Manuscript Figure 7K**). Our data suggest that macrophages from MA-administrated WT mice can functionally contribute to the intestinal I/R injury. We have added these results to the revised manuscript (Results, Line 322).

Author Response Figure 7 (Manuscript Figure 7H-K). (A) Schematic illustration of the induction protocols for macrophages adoptive transfer. (B) H & E staining of small intestinal tissue from mice 36 h after induction of macrophages adoptive transfer and quantification of the small intestinal pathology score (n = 3-4). (C) Quantification of the area of Occludin, ZO-1 immunohistochemical staining (n = 3-4). (D) ELISA detection of IL-6, IL-1 β and IL-10 production in mice 36 h after macrophages adoptive transfer (n = 3-4).

-The source of l-malic acid and the mechanism leading to its increased secretion from intestinal organoids have not been described and discussed.

Response 4: We thank the reviewer for this suggestion. L-malic acid, an important organic acid produced during various metabolic processes, directly participate in mitochondrial energy metabolism by entering the mitochondria through the cell membrane. The source of L-malic acid is mainly through the following metabolic pathways:

Pathway 1: Oxaloacetate reduction pathway. The synthesis of cytosolic malate involves the reduction of oxaloacetate to L-malic acid, catalyzed by malic acid dehydrogenase (Mol Cell. 2018;69(4):581-593.e7.).

Pathway 2: Tricarboxylic acid cycle (TCA cycle) pathway. In this pathway, L-malate is formed by the hydration of fumarate catalyzed by fumarate hydratase (Life (Basel). 2021;11(1):69.).

Pathway 3: Glyoxylate cycle pathway. Glyoxylate and acetyl-CoA react under the influence of malate synthase to generate L-malic acid (Cell Host Microbe. 2011;10(1):33-43.).

The mechanisms leading to the high metabolic level in intestinal organoids may be due to the three-dimensional structure of intestinal organoids. In contrast to conventional two-dimensional cell culture, intestinal organoids exhibit a three-dimensional structure that closely resembles actual intestinal tissue. This structure facilitates increased cell-cell and cell-matrix interactions, thereby promoting an upregulation of metabolic activity (Am J Physiol Gastrointest Liver Physiol. 2015;309(1):G1-G9.). Moreover, intestinal organoids contain various cell types such as intestinal epithelial cells, goblet cells, stem cells, and Paneth cells, among others. These distinct cell populations mutually interact and collaborate within organoid systems, enhancing mitochondrial activity (Nature. 2017;543(7645):424-427.). Furthermore, cells within organoids can modulate metabolic activity through multiple signaling cascades. For example, mucosal cells have the capacity to release hormones and growth factors that activate cell surface receptors and promote metabolic responses (Cell Commun Signal. 2019;17(1):8.).

We have added description and discussed them in the revised manuscript which reads:

Discussion (Line 413): “This could be attributed to their three-dimensional structure, which promotes increased cell-cell interactions and creates signaling cascades by various cell aggregates^{42, 52}. Consequently, this enhances the activity of the TCA cycle”.

Discussion (Line 422): “The primary source of MA production is predominantly attributed to the oxaloacetate reduction pathway, TCA cycle pathway, and glyoxylate cycle pathway^{33, 55, 56}”.

-The cellular and molecular targets of l-malic acid have not been identified in vivo. The direct effect of l-malic acid should be fully demonstrated in vivo using specific SOCS2 macrophage-deficient mice. In addition, to fully demonstrate the potential clinical applicability of this finding, the effect of L-malic acid could be demonstrated independently of organoid transplantation via pharmacological administration in a preclinical I/R injury model

Response 5: We thank the reviewer for this suggestion. Based on the reviewer’s suggestions, we constructed the SOCS2 knocked-out mice to determine whether MA controls macrophage polarization and intestinal recovery from I/R injury in a SOCS2-dependent manner in vivo. We observed that the pathology scores in MA-administrated SOCS2^{-/-} mice were significantly higher than that in the MA-treated wide type (WT) mice (**Author Response Figure 8A, Manuscript Figure 7A**). Moreover, SOCS2^{-/-} mice developed decreased intestinal barrier integrity (**Author Response Figure 8B, C, Manuscript Figure 7B, C**), secreted more IL-6, IL-1 β and fewer IL-10 (**Author Response Figure 8D, Manuscript Figure 7D**). Additionally, SOCS2-deficient mice impaired the ability to generate Ly6C⁺MHC II⁺ and Ly6C⁻MHC II⁺ macrophages (**Author Response Figure 8E, Manuscript Figure 7E**) and reduced the proportion of M2-like macrophages (**Author Response Figure 8F, G, Manuscript Figure 7F, G**). Taken together, these results indicated that MA stimulation promotes macrophages polarization into M2-like macrophages in a SOCS2 dependent manner. We added these results in the revised manuscript (Results, Line 311).

Author Response Figure 8 (Manuscript Figure 7A-G). (A) H & E staining of small intestinal tissue from mice 36 h after induction of intestinal I/R and quantification of the small intestinal pathology score (n = 3-4). (B) Quantification of the area of Occludin, ZO-1 immunohistochemical staining (n = 3-4). (C) qRT-PCR analysis of Occludin and ZO-1 mRNA in small intestinal tissues from mice 36 h after intestinal I/R (n = 3-4). (D) ELISA detection of IL-6, IL-1 β and IL-10 production in mice 36 h after intestinal I/R (n = 3-4). (E) Frequency of Ly6C⁺MHCII⁻, Ly6C⁺MHCII⁺, and Ly6C⁻MHCII⁺ subsets among CD45⁺CD11b⁺ cells in the small intestine of different groups (n = 3-4). (F) Quantification of CD206⁺F4/80⁺CD45⁺CD11b⁺ macrophages (n = 3-4). (G) Representative histograms of CD206 expression in the LP of the small intestine 36 h after intestinal I/R injury.

To further dissect the cellular targets of MA, we conducted macrophages adoptive transfer assay. Please refer to our **response 3** to the comment of the reviewer. These data confirm that SOCS2-containing macrophages contribute to the anti-inflammatory function in the intestinal I/R injury. We added these results in the revised manuscript (Results, Line 311).

Therefore, the results obtained from the comparison between the MA administrated group and the vehicle group under intestinal I/R injury in WT mice, appear to support the reviewer's viewpoint that MA could serve as a novel therapeutic metabolite for intestinal I/R injury treatment. These results are presented in the **Manuscript Figure 7** and added to the Results as mentioned above.

Reviewer #3 (Remarks to the Author):

Overall impression

The manuscript is well written, and clearly illustrates the biological effect of successful organoids transplantation via L-Malic secretion from organoids and its downstream signal cascade. The importance of IL-10 to suppress STAT3 phosphorylation is also well documented with clear results. Not only transplantation system but also culture system using conditioned medium of intestinal organoids are quite unique systems, which would benefit the related field. Particularly this reviewer agreed that organoids transplantation can even modify the systemic immunity based on the content of Figure 2. This part is a quite novel aspect of organoids transplantation, which may be beneficial for a broad range of readers in various research fields. The process to target L-Malic acid is also well-documented and clear.

As one of the experts in the field, this reviewer may have some comments as described below. Please consider take these issues into the manuscripts.

Response to general comment: We thank the reviewer for the positive remarks along with the many helpful suggestions for us to improve the manuscript. Below please find a point-by-point response to the comment. Specifically:

- 1) To improve data clarity, we have increased the magnification of some images.
- 2) We have added more discussion on the reason of higher engraftment rate in organoids compared to sorted Lgr5 cells or crypts.
- 3) We have corrected the typographical errors and made the necessary revisions.

Comments

1. In page 3 in the introduction, the manuscript explains about organoids transplantation citing some of the important references. This reviewer thinks that the following paper might be a candidate of reference here. Satoshi Watanabe et al, Nature Protocols, 2022. Please consider whether this citation is correct one in references.

Response 1: We thank the reviewer for pointing out that some necessary references in the manuscript are missing. Indeed, we have been followed series of studies by Shiro Yui et al., since they published the first organoids orthotopic transplantation system in vivo. We really appreciate the standard protocol provided by their team in the organoids orthotopic transplantation field. We have included this important reference in the Discussion section which reads:

Discussion (Line 379): “Moreover, the standardization protocol for intestinal organoids transplantation and future extensive research will contribute to its application in the treatment of human diseases⁴⁴”.

2. This reviewer thinks that immunofluorescence for Ki67 in Figure 1I and Supplementary Figure 4A is not so clear. Hence the valid quantification supports the conclusion, this reviewer prefers the higher magnification. Please consider which is better.

Response 2: We thank the reviewer for this suggestion. As requested by the Reviewer, in the revised version of the manuscript we have increased the magnification of the images and ensure that the images contain at least five crypts, as shown in (Author Response Figure 9, 10, Manuscript Figure 1I, Supplementary Figure 4A) (from 20x in the previous figure to 40x).

Author Response Figure 9 (Manuscript Figure 1I). Immunofluorescence image of intestinal crypts following Ki-67 staining at indicated groups.

Author Response Figure 10 (Supplemental Figure 4A). Immunofluorescence image of intestinal crypts following Ki-67 staining at indicated groups.

3. In page 7 in the sentence ‘Notably, our results are consistent with previous study of higher engraftment rate of small intestinal organoids grown from ...’ , this reviewer thinks that this is ‘colonic’ instead of ‘small intestinal’ .

Response 3: We thank the reviewer for this suggestion. The reviewer is correct that the published paper we referred to obtained the crypts from colonic tissue. We apologize for this overlook and have corrected ‘small intestinal organoids’ as ‘cultured organoids’ in the results section which reads:

Results (Line 102): “Notably, our results are consistent with previous study ¹¹ of higher engraftment rate of cultured organoids, rather than freshly isolated

crypt or sorted leucine-rich repeat-containing G-protein coupled receptor 5 (LGR5)⁺ stem cells 36 h after intestinal I/R injury (Supplementary Figure 1A-D)".

4. In the same paragraph of page 7, there is a sentence 'we found a scattered distribution of donor organoids in the recipient's small intestinal', but this reviewer thinks this is 'small intestine' rather than 'small intestinal'.

Response 4: We thank the reviewer for this suggestion. We apologize for the typo and have carefully checked and corrected the manuscript as suggested.

5. This reviewer could not find the call for Figure 7.

Response 5: We thank the reviewer for catching this and we apologize for the lack of description on the Figure 8. This figure summarizes what we found neatly and we have added it to the matched description in the discussion section which reads:

Discussion (Line 338): "Here, we report for the first time that small intestine organoids exploit their unique colonization features to influence the local immune microenvironment and promote the self-renewal of ISCs during intestinal I/R injury. Interestingly, organoids secrete MA as a metabolic substrate, which promotes M2 macrophage polarization and restores IL-10 levels in a SOCS2 dependent manner in intestinal I/R injury (Figure 8).

6. In the discussion part, the authors discuss the reason of higher engraftment rate in organoids compared to sorted Lgr5 cells or crypts. This reviewer thinks that amplification of stem cells fraction during in vitro culture

might be one of reasons, and this might be better to include this with appropriate references.

Response 6: We thank the reviewer for this insightful suggestion. During in vitro culture of organoids, the population of stem cells can undergo amplification due to the presence of growth factors and culture conditions that promote their proliferation. This ultimately results in an increased number of stem cells within the organoid, which can enhance the engraftment rate when transplanted into the recipient. (Nature. 2009;459(7244):262-265., Nat Med. 2012;18(4):618-623.). Furthermore, the three-dimensional structure and cell-cell interactions present in organoids provide a more supportive microenvironment for stem cell maintenance and differentiation (Gastroenterology. 2019;157(4):1093-1108.e11.). This microenvironment can enhance the survival and function of transplanted stem cells, thereby further contributing to the higher engraftment rate observed in organoids. As suggested, we have added this discussion in the revised manuscript which reads:

Discussion (Line 369): “In addition, during in vitro culture of organoids, the population of stem cells can undergo amplification due to the presence of growth factors and culture conditions that promote their proliferation⁴³. This ultimately results in an increased number of stem cells within the organoid, which can enhance the engraftment rate when transplanted into the recipient”.

We sincerely thank the editor and reviewers for your warm work and valuable feedback on our manuscript. We have carefully considered all your suggestions and tried our best to improve the manuscript. Your feedback has helped us identify the shortcomings in our current work and we have taken steps to address them. We hope that we have successfully addressed all of your concerns and that the revised manuscript is acceptable for publication. If there are any further modifications that you suggest, we would like very much to modify them and we really appreciate your assistance.

REVIEWERS' COMMENTS

Reviewer #1 (Remarks to the Author):

The authors adequately revised the manuscript and thoroughly responded to the comments of the reviewers. Additional experiments were performed that supported their initial findings.

Reviewer #2 (Remarks to the Author):

The authors used SOCS2 knocked-out mice to evaluate if l-malic acid influences macrophage polarization and intestinal recovery from I/R injury. The authors conformed the role of SOCS2 confirming that l-malic acid fail to induce recovery in SOCS2^{-/-} mice after intestinal ischemia reperfusion (I/R) injury. The SOCS2^{-/-} mice also demonstrated diminished intestinal barrier integrity, produced increased levels of IL-6 and IL-1 β , and reduced IL-10. Furthermore, these mice showed a decreased capacity to generate specific macrophages and a reduced proportion of anti-inflammatory macrophages. Subsequent experiments involved an adoptive transfer assay, revealing that transferring l-malic acid -treated WT macrophages into mice resulted in reduced intestinal I/R injury. Overall, these findings proved that l-malic acid promotes macrophage polarization into anti-inflammatory macrophages through a SOCS2 dependent process.

The experiments presented in the revised manuscript have enhanced the quality of the content and underscore the role of SOC2.

Two minor points need further clarification:

The authors continue to use outdated nomenclature, referring to M2-like macrophages. This should be updated to "anti-inflammatory" or "pro-resolving" macrophages.

The authors state that the SOCS2 knockout mice were "constructed". This phrasing is not suitable. From the "Materials and Methods" section, it's evident that the authors obtained the mice from Zai-Long Chi (Wenzhou Medical University). Additional details regarding the creation of this mouse strain should be included, or a reference to an official repository should be provided.

Reviewer #3 (Remarks to the Author):

As in the original draft, the revised manuscript well describes the novel insight of the therapeutic potential of organoids transplantation by revealing the detailed mechanisms in the modulation of local immune system. The approach to identify the biological benefit of organoids is one of the important issues in the organoids field, and this manuscript would innovate the research in the field of organoids transplantation. This reviewer is convinced that the results presented in this study definitely form the part of it, and that the manuscripts will benefit the broad readers for sure.

The response letter is quite convincing to all of this reviewer's fewer comments. The revised manuscript has more of precise analyses in the molecular basis of their theory in general, and this seems to strengthen the core message with a clear set of new data obtained both in vitro and in vivo. This reviewer has no concerns on the contents.

REVIEWERS' COMMENTS

Reviewer #1 (Remarks to the Author):

The authors adequately revised the manuscript and thoroughly responded to the comments of the reviewers. Additional experiments were performed that supported their initial findings.

Response: We thank the reviewer for the positive remarks.

Reviewer #2 (Remarks to the Author):

The authors used SOCS2 knocked-out mice to evaluate if l-malic acid influences macrophage polarization and intestinal recovery from I/R injury. The authors conformed the role of SOCS2 confirming that l-malic acid fail to induce recovery in SOCS2^{-/-} mice after intestinal ischemia reperfusion (I/R) injury. The SOCS2^{-/-} mice also demonstrated diminished intestinal barrier integrity, produced increased levels of IL-6 and IL-1 β , and reduced IL-10. Furthermore, these mice showed a decreased capacity to generate specific macrophages and a reduced proportion of anti-inflammatory macrophages. Subsequent experiments involved an adoptive transfer assay, revealing that transferring l-malic acid -treated WT

macrophages into mice resulted in reduced intestinal I/R injury. Overall, these findings proved that l-malic acid promotes macrophage polarization into anti-inflammatory macrophages through a SOCS2 dependent process.

The experiments presented in the revised manuscript have enhanced the quality of the content and underscore the role of SOC2.

Two minor points need further clarification:

The authors continue to use outdated nomenclature, referring to M2-like macrophages. This should be updated to "anti-inflammatory" or "pro-resolving" macrophages.

Response: We thank the reviewer for this insightful comment. Currently, a large number of literatures still uses the term "M2-like macrophages" to describe a subtype of macrophages with anti-inflammatory characteristics. These studies typically define and identify M2 macrophages based on the expression of marker molecules, cellular functions, and cytokine production. For example, Wang et al. (J Hepatol. 2022 Aug;77(2):312-325) utilized immunofluorescence and flow cytometry to detect and identify the expression of M1 macrophages in livers of patients with non-alcoholic steatohepatitis. They further explored that macrophage Xbp1 deficiency inhibits M1 polarization and promotes M2 polarization, thus alleviating the progression of non-alcoholic steatohepatitis. The researchers confirmed the important role of Xbp1 knockout M2 macrophages in maintaining tissue homeostasis. Furthermore, Kim et al. (Cell Host Microbe. 2023 Jun 14;31(6):1021-1037) found that gut-associated Akkermansia muciniphila secretes threonyl-tRNA synthetase triggering M2 macrophage polarization, thereby increasing IL-10 and attenuating colitis in mice. M2 macrophages exhibit characteristics of anti-inflammation and immune regulation, playing a crucial role in the inflammatory response. These studies

suggest that despite proposals to update this term, the expression pattern of M2 macrophages is still widely accepted and used, and contributes to our understanding of their role in inflammation and disease processes.

The authors state that the SOCS2 knockout mice were "constructed". This phrasing is not suitable. From the "Materials and Methods" section, it's evident that the authors obtained the mice from Zai-Long Chi (Wenzhou Medical University). Additional details regarding the creation of this mouse strain should be included, or a reference to an official repository should be provided.

Response: We thank the reviewer for this note. We revised the word "constructed" to the more appropriate phrase "carry out an experiment" in order to make the description more suitable. SOCS2^{-/-} mice were generated from C57BL/6 mice by using clustered regularly interspaced short palindromic repeats (CRISPR)/CRISPR-associated 9 (Cas9)-mediated genome engineering technology to delete the fragment of exon 3 by using gRNA1 (TTG GCA GTC GTT TTT CTA GT CGG) and gRNA2 (ATT CAG CTA AAA CTA CCT AA GGG) by the Cyagen Biosciences (Guangzhou, China). Additional details regarding the creation of this mouse strain were added to the manuscript as in the revised manuscript:

Results (Line 310): "Furthermore, we carried out an experiment in the SOCS2 knocked-out mice to determine whether MA regulates macrophage polarization and intestinal recovery from I/R injury in a SOCS2-dependent manner in vivo".

Materials and Methods (Line 478): "SOCS2^{-/-} mice in the C57BL/6 genetic background were generated by using the CRISPR/Cas9-mediated genome engineering technology. The deletion of the exon 3 fragment was carried out using gRNA1 (TTG GCA GTC GTT TTT CTA GT CGG) and gRNA2 (ATT CAG CTA AAA CTA CCT AA GGG) generated by Cyagen Biosciences (Guangzhou, China)".

Reviewer #3 (Remarks to the Author):

As in the original draft, the revised manuscript well describes the novel insight of the therapeutic potential of organoids transplantation by revealing the detailed mechanisms in the modulation of local immune system. The approach to identify the biological benefit of organoids is one of the important issues in the organoids field, and this manuscript would innovate the research in the field of organoids transplantation. This reviewer is convinced that the results presented in this study definitely form the part of it, and that the manuscripts will benefit the broad readers for sure.

The response letter is quite convincing to all of this reviewer's fewer comments. The revised manuscript has more of precise analyses in the molecular basis of their theory in general, and this seems to strengthen the core message with a clear set of new data obtained both in vitro and in vivo. This reviewer has no concerns on the contents.

Response: We thank the reviewer for the favorable recommendation.